

# A new plesiosaurian from the Jurassic–Cretaceous transitional interval of the Slottsmøya Member (Volgian), with insights into the cranial anatomy of cryptoclidids using computed tomography

Aubrey Jane Roberts[1,2], Patrick S. Druckenmiller[3,4],
Benoit Cordonnier[5], Lene L. Delsett[6] and Jørn H. Hurum[6]

[1] The Natural History Museum, London, UK
[2] The National Oceanography Centre, University of Southampton, Southampton, Hampshire, UK
[3] University of Alaska Museum, Fairbanks, AK, USA
[4] Department of Geoscience, University of Alaska, Fairbanks, AK, USA
[5] Physics of Geological Processes, Institute of Geosciences, University of Oslo, Oslo, Norway
[6] The Natural History Museum, University of Oslo, Oslo, Norway

Corresponding author
Aubrey Jane Roberts,
aubrey.roberts@nhm.ac.uk

## ABSTRACT

Cryptoclidids are a major clade of plesiosauromorph plesiosaurians best known from the Middle—Late Jurassic, but little is known regarding their turnover into the Early Cretaceous. Of the known cryptoclidid genera, most preserve only a limited amount of cranial material and of these *Cryptoclidus eurymerus*, displays the most complete, but compressed cranium. Thus, the lack of knowledge of the cranial anatomy of this group may hinder the understanding of phylogenetic interrelationships, which are currently predominantly based on postcranial data. Here we present a nearly complete adult cryptoclidid specimen (PMO 224.248) representing a new genus and species *Ophthalmothule cryostea* gen et sp. nov., from the latest Jurassic to earliest Cretaceous part of the Slottsmøya Member, of central Spitsbergen. The holotype material preserves a complete cranium, partial mandible, complete and articulated cervical, pectoral and anterior to middle dorsal series, along with the pectoral girdle and anterior humeri. High resolution microcomputed tomography reveals new data on the cranial anatomy of this cryptoclidid, including new internal features of the braincase and palate that are observed in other cryptoclidids. A phylogenetic analysis incorporating new characters reveals a novel tree topology for Cryptoclididae and particularly within the subfamily Colymbosaurinae. These results show that at least two cryptoclidid lineages were present in the Boreal Region during the latest Jurassic at middle to high latitudes.

## INTRODUCTION

Plesiosauria is a clade of secondarily aquatic reptiles that predominantly inhabited marine environments during the Mesozoic Era. During the Jurassic, the plesiosaurian fossil record reveals a worldwide distribution and high level of morphological disparity

(*Benson, Evans & Druckenmiller, 2012*). As with many other marine reptile groups, plesiosaur taxonomic diversity was heavily affected by eustatic sea-level changes during the Jurassic–Cretaceous transition (*Tennant, Mannion & Upchurch, 2016*), with the decline and replacement of some Jurassic clades by Xenopsaria (*Benson & Druckenmiller, 2014*). The Middle Jurassic–Early Cretaceous plesiosauroid family Cryptoclididae, is a species-rich clade primarily known from the Northern Hemisphere. The majority of the specimens derive from the Oxford and Kimmeridge Clay Formations of the UK. The recent recovery and description of numerous cryptoclidid specimens from the Slottsmøya Member Lagerstätte of the Agardhfjellet Formation (central Spitsbergen), now constitute a major component of overall Boreal plesiosaurian richness from the Tithonian–Berriasian interval (*Benson & Druckenmiller, 2014*; *Roberts et al., 2017*).

Cryptoclididae includes the subclade Colymbosaurinae *Benson & Bowdler (2014)* and another clade yet to be formally named. Colymbosaurinae previously included all the described plesiosauroid taxa from the Slottsmøya Member (*Djupedalia engeri*, *Spitrasaurus wensaasi*, *S. larseni*, *Colymbosaurus svalbardensis*), in addition to *Abyssosaurus nataliae*, *Pantosaurus striatus* and *Colymbosaurus megadeirus* (*Benson & Bowdler, 2014*). As noted by *Benson & Bowdler (2014)*, the cranial anatomy of cryptoclidids is poorly known and thus the diagnosis of Colymbosaurinae is based exclusively on postcranial characters. The new specimen described here, PMO 224.248, is significant in that it preserves a nearly complete cranium in association with a partial postcranium and represents a new genus and species of cryptoclidid plesiosaurian, *Ophthalmothule cryostea*. PMO 224.248 was excavated from Mt. Wiman in 2012 out the Slottsmøya Member of the Agardhfjellet Formation, from a part of the unit section encompassing the Jurassic–Cretaceous boundary. The specimen represents the fourth and youngest cryptoclidid genus described from the Slottsmøya Member and based on micro computed tomography (μCT) imaging, adds significant new data on the cranial anatomy of cryptoclidids and plesiosaurians in general. These data contribute to a revised phylogenetic hypothesis of cryptoclidids and shed light on plesiosaurian diversity at or near the Jurassic–Cretaceous boundary.

## Geological setting

The Agardhfjellet Formation encompasses a thick succession of Middle Jurassic to Lower Cretaceous sedimentary rocks. The formation comprises four members; Oppdalen Member, Lardyfjellet Member, Oppdalsåta Member and the Slottsmøya Member. The Slottsmøya Member (Volgian) consists of dark-grey to black silty mudstone, which is often weathered into paper shale. There are discontinuous silty beds, with siderite concretions, in addition to siderite and dolomite interbeds. The Slottsmøya Member is overlain by the Lower Cretaceous Myklegardfjellet Bed, which defines the base of the Rurikfjellet Formation (Fig. 1A; *Dypvik et al., 1991*).

The Slottsmøya Member was deposited in an open marine environment under dysoxic conditions (*Collignon & Hammer, 2012*). These marine deposits represent deposition from the upper Tithonian (uppermost Jurassic) to the lower Berriasian (lowermost

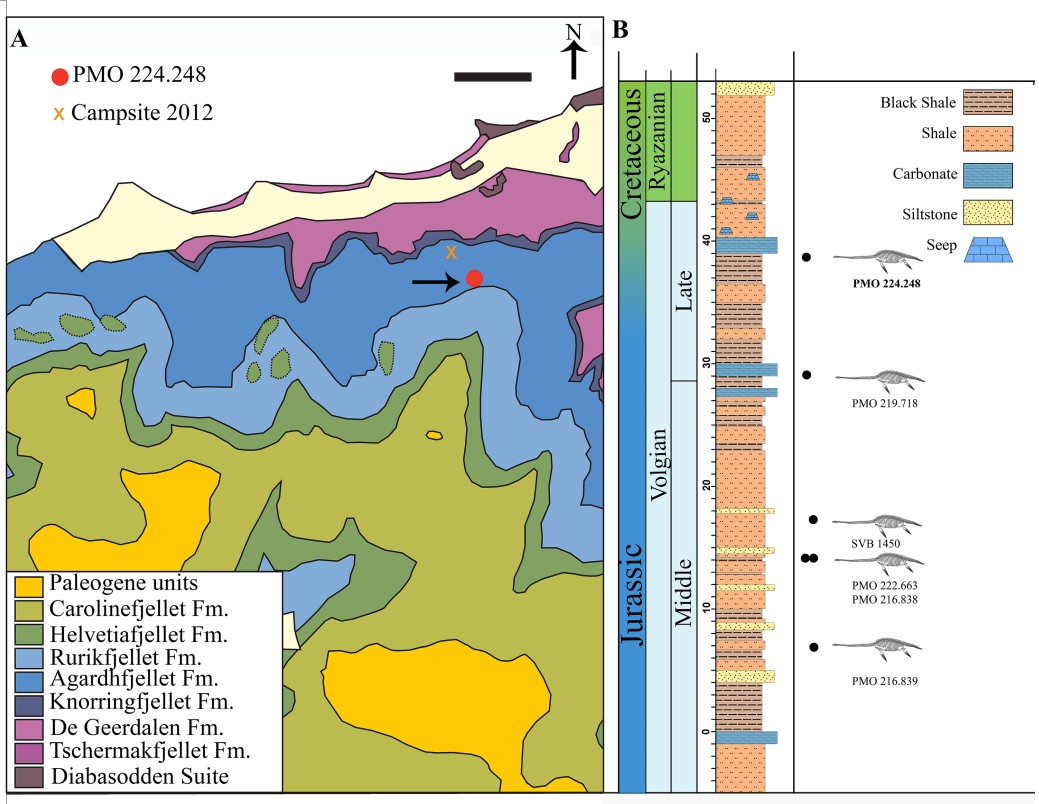

**Figure 1 Map of locality and stratigraphy of the Upper Jurassic (middle Volgian) to the Early Cretaceous part of the Slottsmøya member of the Agardhfjellet Formation (the lowest unit not included, see text) with described cryptoclidid positions.** (A) Geological map of the field site in central Spitsbergen. The arrow points to the excavation site of PMO 224.248 (Modified from *Dallmann et al. (2001)* and *Hurum et al. (2012)*). Scale bar equals 1 km. (B) The stratigraphic position of PMO 224.248 (in bold) in relation to the other described cryptoclidids specimens (PMO 219.718—*Spitrasaurus wensaasi*; PMO 222.663, PMO 216.838—*Colymbosaurus svalbardensis*; PMO 216.839—*Djupedalia engeri*; SVB 1450—*Spitrasaurus larseni*). Note the uncertain position of the Jurassic–Cretaceous boundary (Modified from *Delsett et al. (2016)*).

Cretaceous; *Hammer, Collignon & Nakrem, 2012*). The specimen (PMO 224.248) described here derives from the upper section of the Member (at 38.5 m, estimated from the Janusfjellet log), which is increasingly condensed up section, making it difficult to precisely determine if this specimen derives from the uppermost Jurassic or lowermost Cretaceous (Fig. 1B). However, the seeps overlying the specimen in the stratigraphy are determined to be from the Early Cretaceous (*Hryniewicz et al., 2012*).

# MATERIALS AND METHODS

## Measurements

Measurements were taken using a calliper or tape measure for longer measurements (>15 cm), these are available in the Supplemental Information (Tables S1–S4). For some of the braincase elements that were obscured by matrix or another element, measurements were taken using the µCT scan images.
## μCT methodology

The cranium, left mandible, eighth cervical vertebra and possible gut contents were scanned using μ-computed tomography at the University of Oslo Natural History Museum (Økern Campus). Figures of the volume renderings of the complete posterior of the skull, lower jaw and 8th cervical vertebra are shown in the Supplemental Materials (Figs. S2 and S6). Due to the size limitations of the scanner, three separate scans of the cranium (posterior, middle and anterior of the cranium), were taken and then merged together to form a single high-resolution scan. For the cranium: a total volume of 7,274,887 cm$^3$ was acquired using a Nikon XT H 225 ST desktop CT scanner, with a spatial resolution equal to a voxel size of 0.0753767 mm$^3$. A 2 mm copper filter was utilised. For each scan, tomographic acquisition was performed under step rotation with an exposure time of 2,000 ms, the beam energy was 180 keV and 3,016 projections were taken over 360°. For the left mandible, two μCT scans were performed and then merged, using the same settings as for the cranium. These consisted of 1,583 projections taken over 360°, with an exposure time of 1,000 ms. The 8th cervical vertebra was scanned with 3,016 projections taken over 360°, with 1,000 ms exposure, with the same settings as for the cranium.

Manual segmentation of the braincase was performed with the 3D analysis software Aviso Fire (V. 8.1) and Fiji (ImageJ) at the University of Southampton μ-vis (Muvis) Digital Visualisation Laboratory. The automatic segmentation of the complete cranium was pre-processed with a growing algorithm developed by CB in MATLAB (V. 2016b). This eliminated some of the surrounding and internal matrix from the volume rendering using differences in density. A video of the complete volume rendered skull is available in the Supplemental Information. The complete raw data is available for download from Morphosaurce.org (P774).

## Permits

The following permits were given by the Governor of Svalbard for the University of Oslo Natural History Museum excavations in 2007, 2010, 2011 and 2012: 2006/00528-13; RIS ID 3707; RIS ID: 4760 and 2006/00528-39.

## Nomenclatural acts

The electronic version of this article in Portable Document Format will represent a published work according to the International Commission on Zoological Nomenclature (ICZN) and hence the new names contained in the electronic version are effectively published under that Code from the electronic edition alone. This published work and the nomenclatural acts it contains have been registered in ZooBank, the online registration system for the ICZN. The ZooBank LSIDs (Life Science Identifiers) can be resolved and the associated information viewed through any standard web browser by appending the LSID to the prefix http://zoobank.org/. The LSID for this publication is: (LSID urn:lsid: zoobank.org:pub:3578E578-4724-45FE-8CEE-C075D5C54F34). The online version of this work is archived and available from the following digital repositories: PeerJ, PubMed Central and CLOCKSS.

# SYSTEMATIC PALAEONTOLOGY

Sauropterygia Owen, 1860
Plesiosauria de Blainville, 1835
Plesiosauroidea Welles, 1934
Cryptoclididae Williston, 1925
*Ophthalmothule* gen. nov.
LSID urn:lsid:zoobank.org:act:63110850-0CAC-4DBA-99C2-7AC3B6B926DB
Diagnosis as for the species
*Ophthalmothule cryostea* sp. nov.

LSID urn:lsid:zoobank.org:act:97CEBF5F-58FE-472F-AFA4-9C00E37BB834
(Figures 2–19)

**Holotype:** PMO 224.248

**Occurrence:** The holotype specimen PMO 224.248 was excavated from the north-facing slopes of Wimanfjellet (Mt. Wiman), from the upper part of the Slottsmøya Member, Agardhfjellet Formation, central Spitsbergen: GPS coordinates UTM 33X E523620 N8696396 (Fig. 1). The specimen was located 38.5 m above the yellow storm deposit marker bed (0 m in log), and is late Volgian (latest Tithonian/ early Berriasian) in age.

**Etymology:** *Ophthalmothule. Ophthalmo*, meaning eye. *Thule* is a term used for the northern-most region of the world. Together they make "North eye". Species name, *cryostea*, meaning "frozen bones".

## Differential diagnosis

A moderately sized cryptoclidid plesiosaur (estimated body length of 5.0–5.5 m), possessing the following autapomorphies unique among Cryptoclididae (*) and unique character combinations: premaxilla bears 6 alveoli (5 in *Tricleidus seeleyi* and *Muraenosaurus leedsii*); medial process of premaxilla terminates anterior to the posterior margin of external naris (*); maxilla estimated to contain a similar number of alveoli (>16) as in in *Cryptoclidus eurymerus* (18) and *Tricleidus seeleyi* (15); frontal twice as anteroposteriorly long as parietal (subequal or shorter in *Cryptoclidus eurymerus* and *M. leedsii*); frontal participates in the medial and posterior margins of the external naris (participates posteriorly in *M. leedsii*); presence of an interfrontal vacuity (absent in *M. leedsii*); dorsoventrally low but mediolaterally narrow sagittal crest (flat and mediolaterally broad in *Kimmerosaurus langhami*); quadrate articulates anterolaterally to the pterygoid (posteromedially in *Tricleidus seeleyi* and *M. leedsii*); lateral cotyle of quadrate larger than medial cotyle (reversed in *S. larseni*); basioccipital tubera mediolaterally broad and dorsoventrally flattened (circular in *K. langhami* and *Cryptoclidus eurymerus*); basioccipital tubera triangular in ventral view, following the anteromedial process of pterygoid anteriorly (cylindrical with finished bone anteriorly in

*K. langhami*); exoccipital does not contribute to occipital condyle (contributes in
*K. langhami* and *Cryptoclidus eurymerus*); posteromedian ridge on supraoccipital absent
(present in *K. langhami* and *M. leedsii*); palatine and vomer excludes maxilla from internal
naris (maxilla participates in *M. leedsii*); vomer excluded from anterior interpterygoid
vacuity (participates in *M. leedsii* and *Cryptoclidus eurymerus*); anteromedial process of
pterygoid extends as far as the parabasisphenoid (absent in *Cryptoclidus eurymerus*);
dentary with a mediolaterally extended alveolar surface and with laterally shifted,
labially inclined alveoli (no mediolateral extension and alveoli positioned centrally in
*Tricleidus seeleyi*); deep glenoid facet of the mandible, constituting over half the
dorsoventral height of the mandible (relatively shallow in *Colymbosaurus* spp.,
*Cryptoclidus eurymerus* and *K. langhami*); retroarticular process slightly dorsally inclined
(significantly inclined in *Spitrasaurus larseni*); faint longitudinal ridges on the teeth,
distinct on labial side (distinct on lingual side in *M. leedsii* and *Cryptoclidus eurymerus*;
ridging absent in *K. langhami*); slightly recurved tooth crowns (significantly recurved in
*S. larseni* and *K. langhami*); mamillate hypophyseal eminence is present on the ventral
surface of the atlas (ventral keel or keel-like morphology in *Cryptoclidus eurymerus*,
*M. leedsii* and *Tricleidus seeleyi*); atlantal rib present (absent in *Colymbosaurus
megadeirus*); 50 cervical vertebrae[*] (32 in *Cryptoclidus eurymerus*; 44 in *M. leedsii*; 41 in
*Colymbosaurus megadeirus*; 60 in *Spitrasaurus wensaasi*); cervical centra are slightly
amphicoelous (conspicuously concave in *Djupedalia engeri* and *K. langhami*); cervical
vertebra eight with anteroposteriorly long postzygapophyses, close to the length of
centrum (autapomorphic among Plesiosauria)[*]; anterior-most cervical neural spines low
and posteriorly angled (straight in *K. langhami*); cervical prezygapophyses unfused
anteriorly and fused posteriorly (unfused throughout in *Cryptoclidus eurymerus* and
completely fused in *Spitrasaurus* spp. and *D. engeri*); postzygapophyses fused along the
midline (unfused in posterior-most cervicals in *D. engeri*); lateral ridges present on
mid-posterior cervicals (absent in *Colymbosaurus megadeirus*, *Cryptoclidus eurymerus*,
*D. engeri*, and *Tricleidus seeleyi*); posterior cervical—anterior dorsal ribs with a distinct,
short (<half the rib length) longitudinal ridge along the dorsal surface of the rib[*]; dorsal
vertebral rib facets dorsoventrally taller than wide (circular in *Tatenectes laramiensis*);
dorsal process of scapula short and reduced (tall and extensive in *Abyssosaurus nataliae*
and *D. engeri*); extended anteromedial process of coracoid (reduced in *Colymbosaurus
megadeirus* and *Abyssosaurus nataliae*); humeri significantly larger than femora (femora
larger than humeri in *D. engeri*, subequal in *Colymbosaurus svalbardensis*); sigmoid
humerus in dorsal view, with the proximal end (capitulum) of the humerus angled
anteromedially[*]; humeri with three to four distal articular facets (two in *Cryptoclidus
eurymerus* and *M. leedsii*); radius slightly larger than ulna (anteroposteriorly shorter in
*Colymbosaurus svalbardensis*).

## Taphonomy

The skeleton of PMO 224.248 is well-preserved and fully articulated, with the exception
of the skull, some dorsal vertebrae and distal phalangeal elements (Fig. 2). The carcass had
a ventral landing and the cranium drifted 20 cm from its original position, and

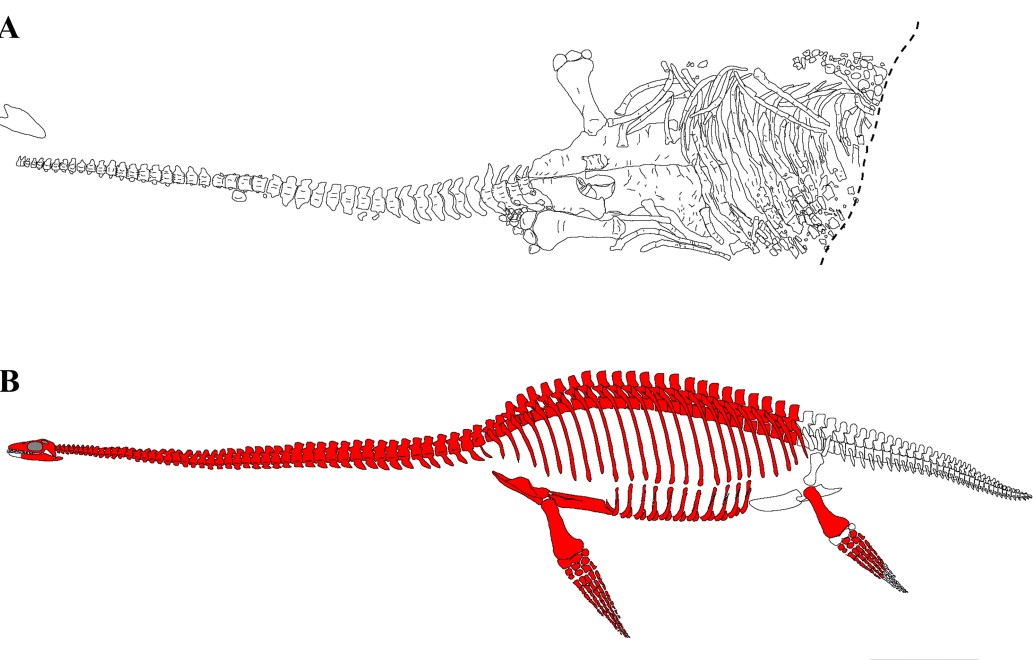

**Figure 2 Quarry map and reconstruction of PMO 224.248.** (A) Drawing from a combination of field and laboratory drawings in ventral view (modified from *Delsett et al. (2016)*); (B) skeletal reconstruction of PMO 224.248, where red indicates preserved elements. Scale bar equals 50 cm. Drawn by Aubrey Jane Roberts.

the majority of the phalanges drifted from the limbs and were not preserved in articulation. The skeleton posterior of the sacral region was eroded, although a partial left hindlimb and femoral fragments were recovered.

The specimen preserves a nearly complete skull, although it is dorsoventrally crushed and damaged in places. The cranium could not be completely prepared from the matrix, as the bones were significantly fractured. The dorsal portions of the braincase were partly disarticulated due to the crushing, with the supraoccipital pushed down into the foramen magnum. The majority of the cervical vertebrae are missing portions of the right cervical rib and neural spines, due to crushing and/or pre-burial erosion. The pectoral and anterior dorsal vertebrae are partly crushed and distorted by the overlying pectoral girdle. Most of the neural arches are missing from the dorsal vertebrae, as these were exposed to post-diagenetic erosional processes.

## Ontogeny

PMO 214.248 is interpreted to be an adult based on its large size and presence of fused neurocentral sutures throughout the preserved vertebral column, in addition to the fusion of the cervical ribs to the centra (*Brown, 1981*). Other indicators of mature stage of are the fusion along the medial facet of the coracoids and well-formed distal facets of the humeri (*Brown, 1981*).

# DESCRIPTION AND COMPARISON WITH OTHER CRYPTOCLIDID TAXA

## Cranium

The temporal fenestrae are conspicuously small relative to the size of the orbits, being approximately 17% of total skull length, whereas the orbit constitutes approximately 29% of total skull length (Figs. 3–5). The tooth row extends about 75% of the total skull length. For selected measurements of the cranium see Supplemental Information (Table S1).

### Premaxilla

The premaxilla of *Ophthalmothule cryostea* forms the majority of the dorsal and lateral surfaces of the rostrum anterior to the orbit (Fig. 3). As with all cryptoclidids (at least those few that preserve cranial material), the rostrum is relatively short (*Andrews, 1910*; *Brown & Cruickshank, 1994*), having a preorbital to total skull length ratio of 0.43. Although this appears high in comparison to other cryptoclidids, it is due to relatively short post orbital region of the skull. Similar to *Muraenosaurus leedsii*, the dorsal surface of the premaxilla is rugose, forming numerous low and sharp crests, but significantly smoother than in *Tricleidus seeleyi* (*Andrews, 1910*). The premaxillae form a narrow ridge that extends along most of the rostral midline. The anterior portion of the premaxilla-maxilla suture is visible in dorsal view, extending from the rostral margin towards the external naris. The external nares of *O. cryostea* are positioned directly anterior to the orbital margin, being relatively more posterodorsally placed than in *M. leedsii* (*Andrews, 1910*; *Brown, 1981*). In *O. cryostea*, the dorsomedial process of the premaxilla, forms the anterior and anteromedial borders of the external naris. However, the process terminates anterior to the posterior margin of the external naris, representing an autapomorphy of this taxon among cryptoclidids. In *Cryptoclidus eurymerus* the morphology of this region is ambiguous in PETMG R.283.412, due to the preservation of the specimen; however, it has been reconstructed with the premaxillae excluded from most of the medial margin of the external naris by the frontal (*Brown & Cruickshank, 1994*). This morphology is not homologous with the condition where the anterior flange of the frontal, excludes the premaxilla from the external naris as in some rhomaleosaurids (*Smith & Benson, 2014*). In *M. leedsii* and *Tricleidus seeleyi*, the premaxillae form the medial margin of the external nares and either terminate at or continue past the posterior margin of the external nares (*Andrews, 1910*; *Brown, 1981*). The μCT scans of PMO 224.248 confirm that the premaxilla overlaps the anterior portion of the frontal and that this sutural contact lies approximately in line with the external naris, as in *Cryptoclidus eurymerus* (PETMG R.283.412; *Andrews, 1910*; *Brown & Cruickshank, 1994*). Furthermore, in *O. cryostea* the premaxilla-frontal suture is embayed anteriorly along the midline with the longest dimension of the premaxilla occurring in the parasagittal plane. In contrast, the dorsomedial process of the premaxilla taper posteriorly along the midline in *C. eurymerus* and *M. leedsii* (*Andrews, 1910*; *Brown & Cruickshank, 1994*).

Among cryptoclidids the number of premaxillary alveoli varies between 5 and 8 on each side (*Brown, 1981*). Based on μCT data (Fig. 4B), the premaxilla of *Ophthalmothule cryostea* has a total of six alveoli on each side, the same number as *Cryptoclidus eurymerus*

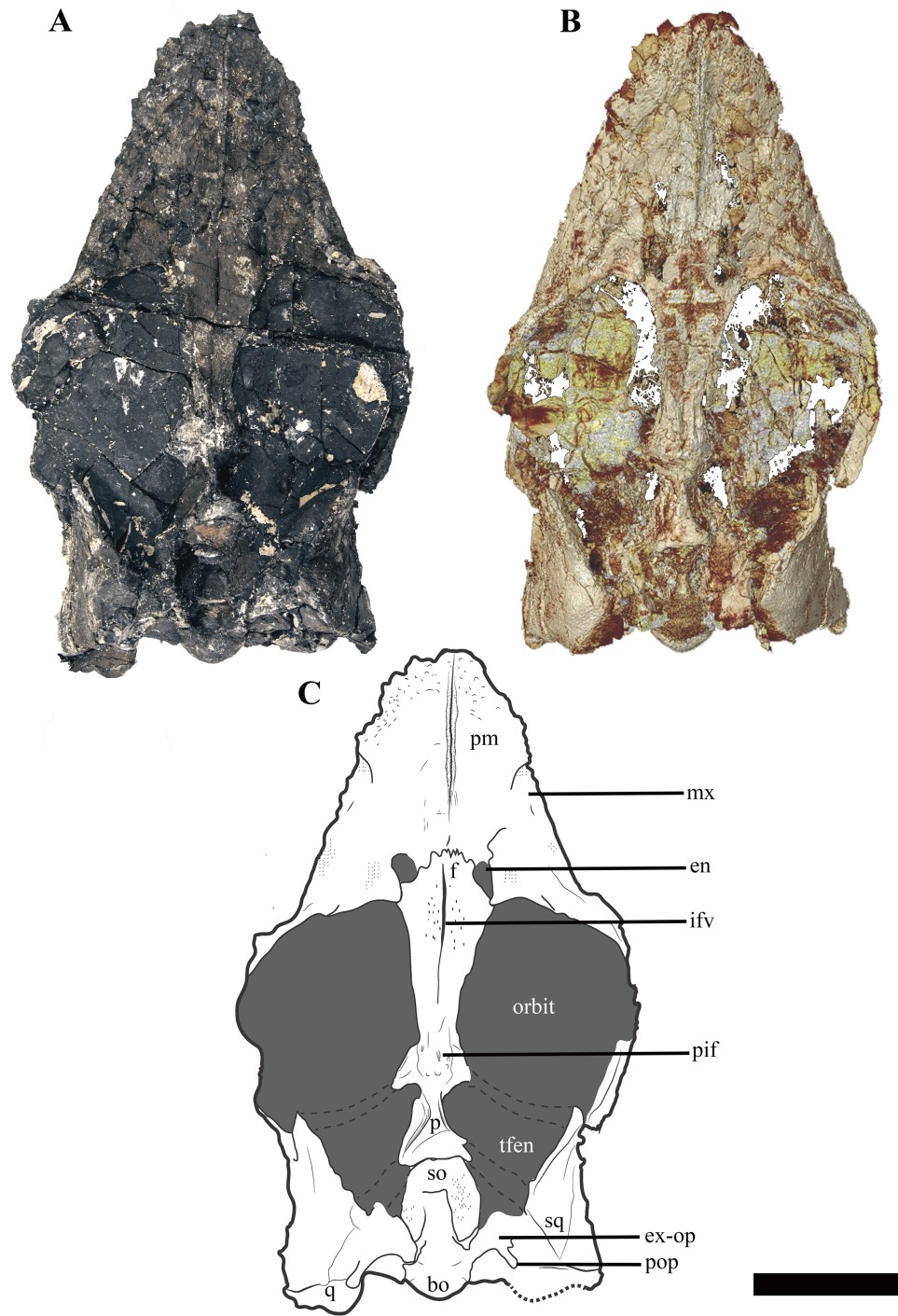

**Figure 3 The cranium of *Ophthalmothule cryostea*, PMO 224.248 in dorsal view.** (A) Photo of PMO 224.248, (B) μCT reconstruction and (C) interpretation. Abbreviations: bo, basioccipital; en, external naris; ex-op, exoccipital-opisthotic; f, frontal; ifv, interfrontal vacuity; mx, maxilla; p, parietal; pif, pineal foramen; pm, premaxilla; pop, paraoccipital process; q, quadrate; so, supraoccipital; sq, squamosal; tfen, temporal fenestra. Scale bar equals 5 cm. Photograph and reconstruction by Aubrey Jane Roberts.

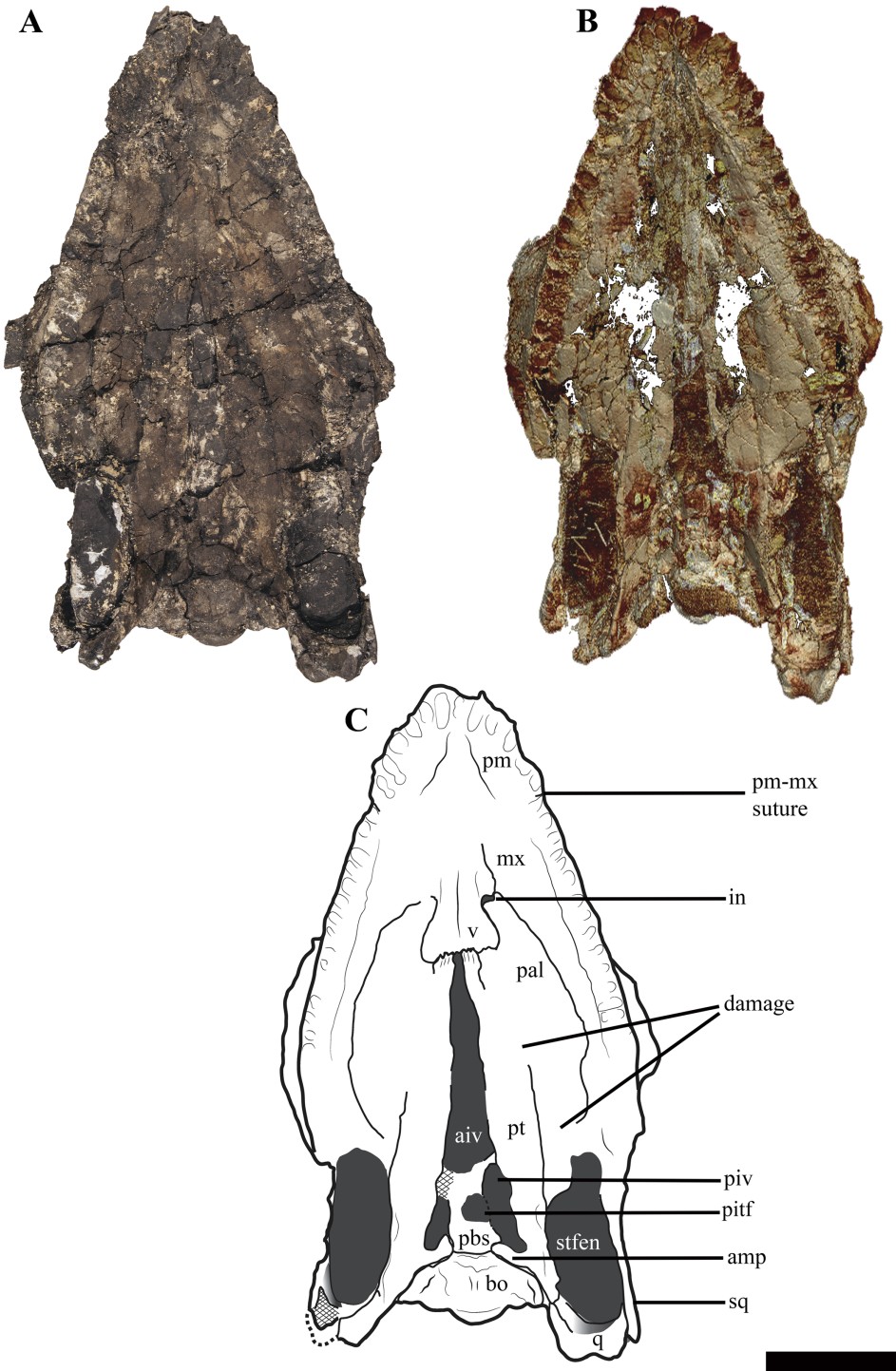

**Figure 4 The cranium of *Ophthalmothule cryostea*, PMO 224.248 in palatal view.** (A) Photo, (B) µCT reconstruction and (C) interpretation. Abbreviations: aiv, anterior interpterygoid vacuity; bo, basioccipital; in, internal naris; mx, maxilla; pal, palatine; pbs, parabasisphenoid; pitf, pituitary fossa; piv, posterior interpterygoid vacuity; pm, premaxilla; pt, pterygoid; q, quadrate; sq, squamosal; stfen, subtemporal fenestra; v, vomer. Scale bar equals 5 cm. Photograph and reconstruction by Aubrey Jane Roberts.

(*Brown & Cruickshank, 1994*), but greater than that observed in *Muraenosaurus leedsii*, *Tricleidus seeleyi* and *Vinialesaurus caroli* (five: *Brown, 1981*; *Gasparini, Bardet & Iturralde-Vinent, 2002*) and less than that suggested for *Kimmerosaurus langhami* (minimum eight: *Brown, 1981*). The first and sixth alveoli are noticeably smaller in all dimensions than the other premaxillary alveoli, which are otherwise similar in size. The premaxillary–maxillary suture is visible in ventral view, just posterior to the sixth alveolus (Figs. 4 and 5).

### Maxilla

The maxilla of *Ophthalmothule cryostea* forms the ventral rim and most of the anterior margin of the orbit. In dorsal view, the prefrontal-maxilla suture is equivocal due to significant breakage in this region. The lateral surface of the maxilla is lightly pitted and rugose, but not to the same extent as the premaxilla. In ventral view the alveoli are partially obscured by matrix, but can be counted using μCT images, showing a maximum of 16 alveoli on both sides when taking damage into account. This is less than *Cryptoclidus eurymerus* (18; *Brown, 1981*), but similar to other Oxford Clay Formation cryptoclidids (16 in *Muraenosaurus leedsii*; 15 in *Tricleidus seeleyi*; *Brown, 1981*). The maxillary alveoli vary only slightly in size and morphology, with the larger labiolingually expanded alveoli located more anteriorly and smaller, more rounded alveoli posteriorly. We interpret the slight asymmetry regarding maxillary alveolus size present in *O. cryostea* to variation in tooth replacement stage. This morphology differs from *Tricleidus seeleyi*, where clear anisodonty is present (*Brown, 1981*). As in *Cryptoclidus eurymerus* (PETMG R.283.412), the posterior extent of the maxillary tooth row in *O. cryostea* terminates in line with the position of the postorbital bar and is positioned considerably higher than the glenoid fossa in lateral view.

Ventrally, the maxilla approaches and nearly contributes to the margin of the internal naris, but is excluded by the palatine-vomer contact, similar to *Cryptoclidus eurymerus* (*Andrews, 1910*). This morphology differs from *Muraenosaurus leedsii* where the premaxilla and maxilla contribute to the anterior and lateral margins of the internal naris respectively (*Andrews, 1910*; *Brown & Cruickshank, 1994*).

### Prefrontal

The region anterior to the orbit in PMO 224.248 is difficult to interpret, due to poor preservation. There are two possible sutures that could represent the lateral and medial margins of a prefrontal, with the position of these sutures confirmed by differences in bone orientation using μCT images (Fig. S1). Using these margins, the prefrontal would be constrained to a small wedge-shaped section directly anterior to the orbital rim, separated from the external naris by a dorsal process of the maxilla. The element is thickened along the orbital margin and posterodorsally overlaps the frontal in a pointed process. The prefrontal is rarely described in Callovian cryptoclidids, with the exception of *Muraenosaurus leedsii* (*Andrews, 1910*; *Brown, 1981*). This has been attributed to either poor preservation of this area in most specimens, or because the element is indiscernible due to fusion with the maxilla (*Brown, 1981*; P.S. Druckenmiller, 2016, personal observation).

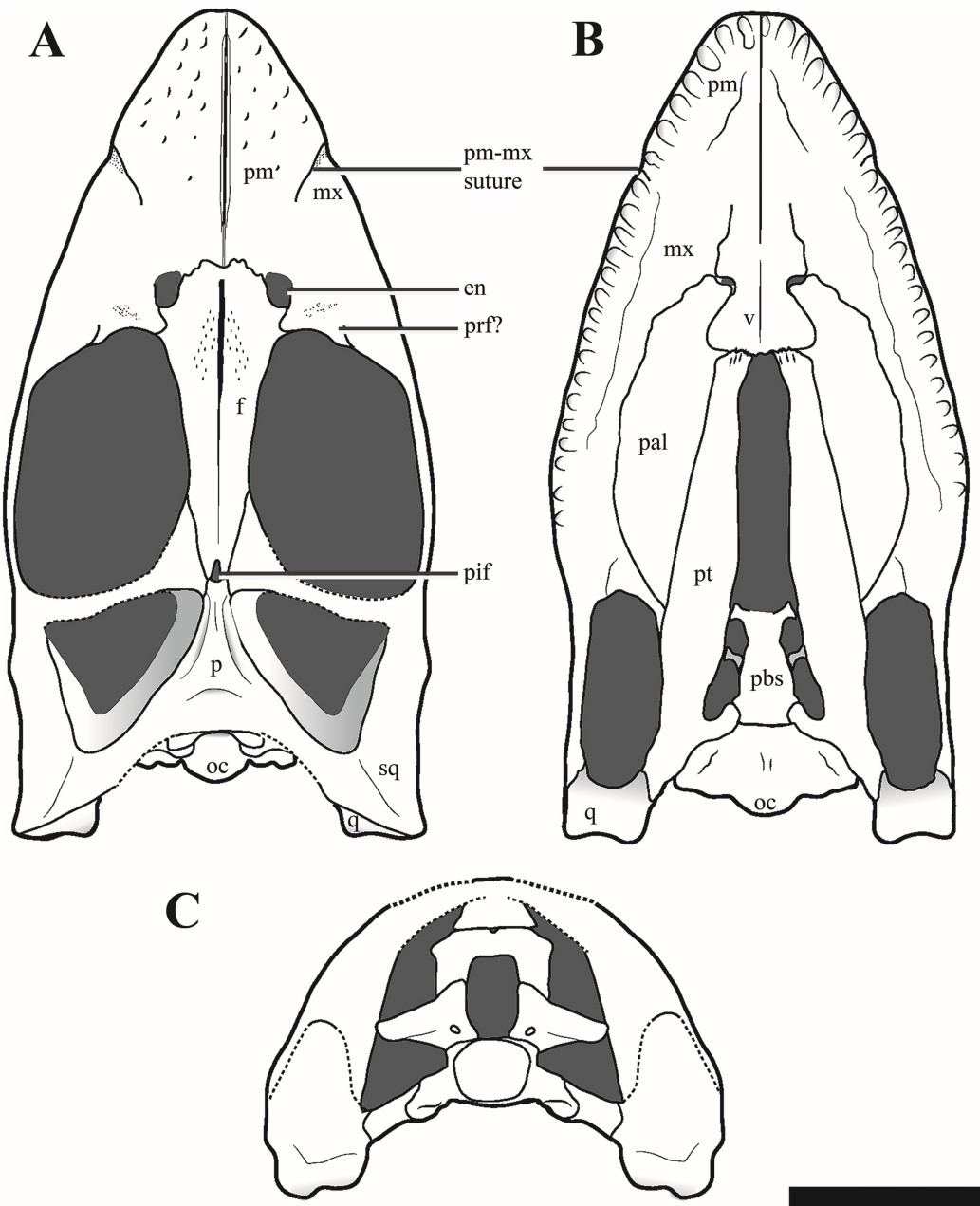

**Figure 5 Reconstructions of the cranium of *Ophthalmothule cryostea*, PMO 224.248.** (A) Dorsal, (B) palatal and (C) posterior views. Abbreviations: aiv, anterior interpterygoid vacuity; ba, basal articulation; bo, basioccipital; boc, basioccipital condyle; bot, basioccipital tuber; en, external naris; ex-op, exoccipital-opisthotic; f, frontal; ifv, interfrontal vacuity; in, internal naris; mx, maxilla; p, parietal; pal, palatine; pbs, parabasispenoid; pif, pineal foramen; pitf, pituitary fossa; piv, posterior interpterygoid vacuity; pm, premaxilla; pt, pterygoid; q, quadrate; so, supraoccipital; sq, squamosal; stfen, subtemporal fenestra; tfen, temporal fenestra; v, vomer. Scale bar equals 5 cm. Drawing by Aubrey Jane Roberts.

### Frontal

In *Ophthalmothule cryostea*, the anteroposterior length (measured along the midline) of the frontal is 2.3 times longer than the length of the parietal, whereas in other taxa the

relative lengths are nearly the same (=~0.9 *Cryptoclidus eurymerus* PETMG R.283.41; =~1 *Cryptoclidus eurymerus* NHMUK R2860; =~0.8 *Muraenosaurus leedsii* using *Andrews (1910)*; =~1.4 *Tricleidus seeleyi* NHMUK R3539). Anteriorly, the frontal participates in the medial and posterior margins of the external naris. In *M. leedsii*, the frontal participates in the margin of the external naris, but lacks the same degree of anterior extension seen in *Ophthalmothule cryostea* (*Andrews, 1910*). As in *M. leedsii*, the greatest mediolateral width of the frontal occurs directly in line with the anterior margin of the orbit, in contrast to *Cryptoclidus eurymerus* where this occurs more posteriorly along the middle orbital margin (*Andrews, 1910*; *Brown & Cruickshank, 1994*). At the point of articulation with the parietal the mediolateral width of the element is roughly a third of the maximum mediolateral width. Along the dorsal margin of the orbit, the frontal has a concave margin, differing considerably from the straight frontal margin of *Muraenosaurus* specimens (*M. leedsii*; NHMUK R.2678) and from *Cryptoclidus eurymerus*, where it is convex (*Cryptoclidus eurymerus*; PETMG R.283.412). The relationships of the postfrontal and postorbital to the skull roof are not preserved.

The frontals form a long, slit-like interfrontal vacuity at their mid-length. At their anterior- and posterior-most ends (adjoining the premaxillary and parietal contacts, respectively), the frontals are fully in contact and fused along the midline. However, along the remainder of their length they lack a firm midline contact and enclose a narrow and elongate slit-like opening, referred to here as the interfrontal vacuity. This vacuity is not homologous to the 'frontal foramen' observed near the anterior margin of the frontal in some polycotylids, which is not located along the midline (*Carpenter, 1996*; *Fischer et al., 2018*). However, this vacuity does bear some resemblance to the dorsomedian frontal foramen described in *Brancasaurus brancai* (*Sachs, Hornung & Kear, 2016*). The most conspicuous development of the interfrontal vacuity occurs in the anterior half of the frontals where it can be recognised by having smooth, finished bone medially and a slight concavity following the midline. An interfrontal vacuity is also clearly present in several other cryptoclidids (e.g. *Kimmerosaurus langhami*; *Tatenectes laramiensis*; *Tricleidus seeleyi*; possibly *Cryptoclidus eurymerus*) and could represent a new synapomorphy for a subclade of Cryptoclididae (see "Discussion"). The dorsal surface of the frontal in PMO 224.248 is generally smooth, but is textured with a few small indentations adjacent to the interfrontal vacuity.

The morphology of the ventral surface of the frontal in *Ophthalmothule cryostea* is visible in the μCT images and tapers medially in cross section. On the ventral surface of the frontal in *Kimmerosaurus langhami*, *Cryptoclidus eurymerus* and *M. leedsii* a trough is present on either side of the interfrontal vacuity or frontal midline. In *K. langhami* these are clearly seen, starting posteriorly in line with the pineal foramen and terminate at the preserved anterior end of the frontal (A.J. Roberts, 2015, personal observations, NHMUK R8431). These structures are absent in *Tricleidus seeleyi* (NHMUK R3539) and in *O. cryostea* based on the μCT images.

The frontal-parietal suture is somewhat obscured in dorsal view due to gypsum mineralisation and the presence of rugosities in the anterior portion of the parietal. However, μCT images show that the posterior margin of the frontal interdigitates with the

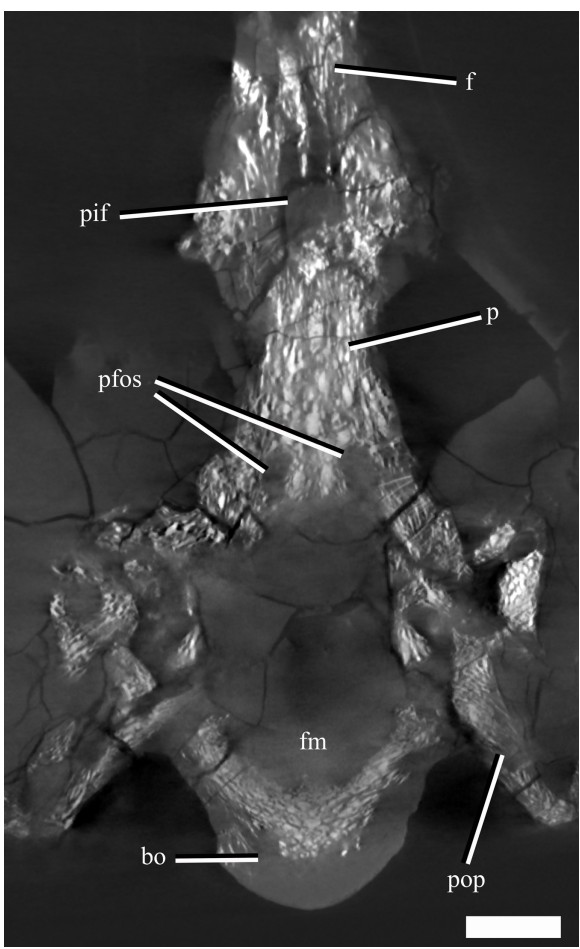

**Figure 6 A μCT slice (cross section) of the posterior part of the skull roof and braincase of** *Ophthalmothule cryostea*, **PMO 224.248, illustrating the parietal fossae and pineal foramen.** Abbreviations: bo, basioccipital; f, frontal; p, parietal; pfos, parietal fossae; pif, pineal foramen; pop, paraoccipital process. Scale bar equals 1 cm.

anterior margin of the parietal and that the frontal envelopes the anterior rim and most of the lateral rims of the pineal foramen (Fig. 6).

### Parietal

The anterior extent of the parietal lies approximately in line to the level of the temporal bar. In dorsal view, the parietal bears a mediolaterally narrow sagittal crest that is slightly flattened dorsally, exhibiting an intermediate condition between the tall and sharp crest seen in *Tricleidus seeleyi*, *Muraenosaurus leedsii* and *Cryptoclidus eurymerus* (*Brown, 1981*; *Brown & Cruickshank, 1994*) and the broad, flat sagittal crest seen in *Kimmerosaurus langhami* (*Brown, 1981*). In lateral view, the apex of the sagittal crest is straight and gently inclines posterodorsally. In contrast, this morphology differs from the dorsally convex sagittal crest seen in *Cryptoclidus eurymerus*. The squamosal-parietal contact is indiscernible.

The μCT images reveal the presence of two large dorsoventrally oriented fossae in the parietals that open onto the posteroventral surface, but do not extend to the dorsal surface

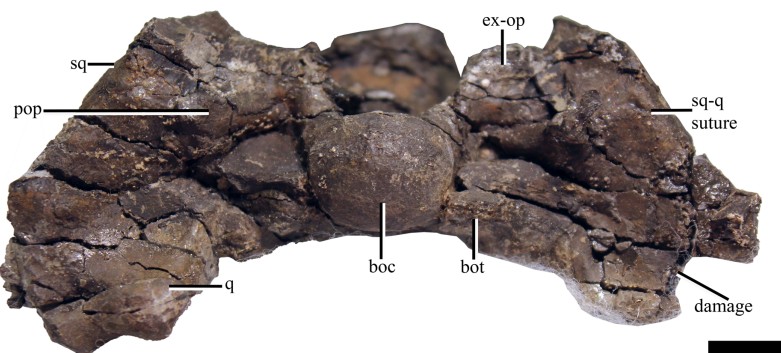

**Figure 7 The cranium of *Ophthalmothule cryostea*, PMO 224.248 in posterior view.** Abbreviations: boc, basioccipital condyle; ex-op, exoccipital opisthotic; q, quadrate; sq, squamosal. Scale bar equals 1 cm. Photograph by Aubrey Jane Roberts.

(Fig. 6). This feature is also present in another undescribed Slottsmøya Member cryptoclidid (PMO 212.662). In *Kimmerosaurus langhami* (NHMUK R.8431), these parietal fossae are likely absent, as they are not visible on the ventral surface (*Brown, 1981*). In *Cryptoclidus eurymerus* (NHMUK R.2860) and *Muraenosaurus leedsii* (NHMUK R.2422) the parietals are partly obscured by the supraoccipital and parietal fossae cannot be identified. Although the function of these fossae is currently unclear, CT scanning of additional specimens could infer whether this is a cryptoclidid feature or more constrained to a specific clade.

### Squamosal

In lateral view, the suspensorium is nearly vertically inclined, although the dorsal half of the squamosal dorsal ramus is inflected abruptly anteriorly (Fig. S2). The squamosal bears a dorsoventrally tall anterior ramus which curves slightly medially, following part of the anterior margin of the temporal fenestra. The ventromedial process of the squamosal is short, extending ventrally to roughly half the dorsoventral length of the quadrate shaft. The dorsal margin of the squamosal-quadrate suture is visible in posterior view, where a small groove is present (Fig. 7). However, this suture could not be located dorsally in μCT scan images (Fig. S2). The relationships between the squamosal and the jugal and postorbital cannot be discerned due to poor preservation in this area.

### Quadrate

Due to a fracture running along the middle of the right quadrate, the left quadrate is better preserved. Similar to *Djupedalia engeri* and *Kimmerosaurus langhami*, the lateral cotyle of the quadrate condyle in *Ophthalmothule cryostea* is slightly larger in anteroposterior length and dorsoventral extent than the medial cotyle (*Brown, Milner & Taylor, 1986*; *Knutsen, Druckenmiller & Hurum, 2012a*). This differs from *Spitrasaurus larseni*, where the opposite state is present (*Knutsen, Druckenmiller & Hurum, 2012b*). There is no indication of a quadrate foramen.

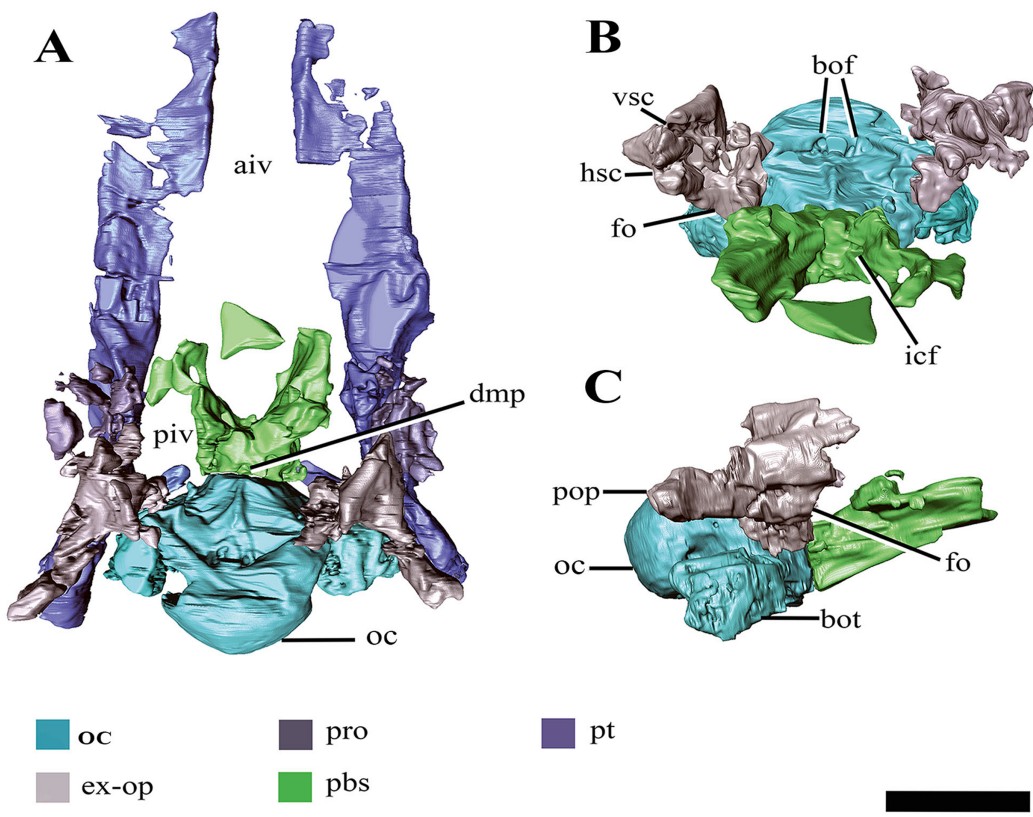

**Figure 8** Surface reconstruction of the braincase and pterygoids of *Ophthalmothule cryostea*, PMO 224.248 using μCT images. In (A) dorsal, (B) anterodorsal and (C) right lateral views. Abbreviations: aiv; anterior interpterygoid vacuity; bo; basioccipital; boc, basioccipital condyle; bof, basioccipital foramina; bot, basioccipital tuber; dmp, dorsal median pit; ex-op; exoccipital-opisthotic; fo; fenestra ovalis; hsc, horizontal semicircular canal; icf, internal carotid foramen; pbs, parabasisphenoid; piv, posterior interpterygoid vacuity; pop, paraoccipital process; pro, prootic; pt, pterygoid; vsc, vertical semicircular canal. Scale bar equals 2 cm.

## Basioccipital

The anterior and dorsal surfaces of the basioccipital of *Ophthalmothule cryostea* are obscured by matrix and other skull elements, but can be described fully using the μCT segmentation (Fig. 8). The occipital condyle lacks both a notochordal pit and a constriction on the dorsal and ventral surfaces; however, a slight constriction is visible on the lateral surfaces. As in *Spitrasaurus larseni*, the exoccipital facets reach, but do not contribute to the occipital condyle (*Knutsen, Druckenmiller & Hurum, 2012b*). In contrast, the exoccipitals in *K. langhami* and in some specimens of *Cryptoclidus eurymerus* form a portion of the condyle (*Andrews, 1910*; *Brown, 1981*). In posterior view, the condyle is mediolaterally wider than dorsoventrally tall. The height-to-width ratio (H/W) of the condyle (~0.82) is comparable to that of *S. larseni* (0.8) and *Kimmerosaurus langhami* (0.85) (*Knutsen, Druckenmiller & Hurum, 2012b*), but differ from *Muraenosaurus leedsii* and *Tricleidus seeleyi* that possess more circular condyles (H/W ~1).

Similar to *Tricleidus seeleyi* and another undescribed Slottsmøya Member cryptoclidid specimen (PMO 212.662), the basioccipital tubera of *Ophthalmothule cryostea* are

dorsoventrally flattened and triangular in general outline as seen in ventral view and their ventral surfaces are gently concave in occipital view (Fig. S2). The entire anterolateral margin of the tubera meet and extend parallel to the pterygoid extending to the basisphenoid margin. This morphology contrasts to the pillar-like (circular in cross section) and laterally-facing tubera in *Cryptoclidus eurymerus* and *Kimmerosaurus langhami* (*Andrews, 1910*; *Brown, 1981*). In addition to this morphology, *K. langhami* displays finished bone along the anterolateral basioccipital margin between the basisphenoid facet and tubera (*Brown, 1981*; *Brown & Cruickshank, 1994*).

The posterior floor of the foramen magnum is visible, forming part of a shallow but mediolaterally broad concavity between the two exoccipital-opisthotics. Anteriorly, there is a low anteroposteriorly oriented ridge that terminates near the contact with the basisphenoid (Fig. 8B). Two paired foramina open on to the dorsal surface of the basioccipital and extend into the body of the element, where they are visible in the μCT images, but terminate before reaching the ventral surface. Similar to *Kimmerosaurus langhami*, the ventral surface of the basioccipital is relatively flat, with a short anteroposteriorly oriented median ridge that terminates at the rim of the anterior margin (*Brown, 1981*).

### Parabasisphenoid

The demarcation between the parasphenoid and basisphenoid of PMO 224.248 is indiscernible. In dorsal view on the posterior margin of the body of the parabasisphenoid, a small fossa is present along the suture with the basioccipital ('dmp,' Fig. 8A). This structure appears homologous with the 'dorsal median pit' described in *Muraenosaurus leedsii* (*Andrews, 1910*), hypothesised to mark the embryonic basicranial fenestra. However, the foramen present in PMO 224.248, is significantly reduced in comparison to *M. leedsii* (*Andrews, 1910*). A deep fossa is present on the anterior margin of the basisphenoid body interpreted to be the pituitary (or hypophyseal) fossa, similar to that described in *Tricleidus seeleyi*, *Kimmerosaurus langhami* and *M. leedsii* (*Andrews, 1910*; *Brown, Milner & Taylor, 1986*). The ventral floor of the pituitary fossa, including parts of both the basisphenoid and parasphenoid (*Andrews, 1910*), is missing likely due to taphonomic loss. In lateral view the internal carotid foramen is visible opening into the pituitary fossa (not visible on right). In dorsal view on the posterior margin of the parabasisphenoid, a small fossa is present along the suture with the basioccipital. In palatal view, the basal articulation of plesiosauroids is often visible in ventral view through the posterior interpterygoid vacuity (*Buchy, Frey & Salisbury, 2006*). In PMO 224.248, the basal articulation is visible on the μCT scans and is better preserved on the left side and positioned dorsally in respect to the rest of the palate.

The parabasisphenoid appears to bear posterolaterally located facets for the anteromedial process of the pterygoid similar to *Tricleidus seeleyi*, but in contrast to *Muraenosaurus leedsii* and *Cryptoclidus eurymerus* where the pterygoid simply articulates to the basioccipital tuber (*Andrews, 1910*; *Brown, 1981*). In *Tricleidus seeleyi*, the pterygoid facets of the parabasisphenoid are circular in outline and are slightly anterolaterally projecting, whereas in PMO 224.248 the facet surface appears uniform and triangular in shape in the μCT images. The presence of a pterygoid facet on the body of a

parabasisphenoid is also present in another Slottsmøya Member cryptoclidid (PMO 212.662) and has been suggested to be present in *Kimmerosaurus langhami* (NHMUK R10042; *Benson & Druckenmiller, 2014*: Appendix S2). That a minimum of three cryptoclidid taxa share this palatal configuration over a long temporal span (Callovian–Volgian), suggests that this morphology is more widespread in cryptoclidids than previously believed.

Anteriorly, the parabasisphenoid is very thin and somewhat damaged, making this region difficult to segment out of the μCT images. The anterior portion (parasphenoid) appears to separate the pterygoids along the midline and forms the entire posterior margin of the anterior interpterygoid vacuity, although it seemingly lacks a projecting cultiform process, like that seen in some polycotylids and basal plesiosaurians (*Buchy, Frey & Salisbury, 2006*; *Carpenter, 1996*; *O'Keefe, 2001*; *Vincent & Benson, 2012*). The anterior margin of the posterior interpterygoid vacuity is formed by a lateral extension of the parabasisphenoid.

### Exoccipital-opisthotic

Both exoccipital-opisthotics are preserved in partial articulation, but are damaged and displaced venterolaterally due to compression. As only the posterior view of these elements is visible on the specimen (Fig. 7), the following description is largely based on μCT images (Fig. 9).

The body of the exoccipital-opisthotic in *Ophthalmothule cryostea* is dorsoventrally taller than mediolaterally wide. In posterior view, the paraoccipital process is visible extending laterally from the body of the element. The cross-sectional shape of the paraoccipital process shaft is dorsoventrally taller than wide. Similar to *Muraenosaurus leedsii*, the length of the paraoccipital process is close to the dorsoventral height of the exoccipital, in contrast to *Tricleidus seeleyi* and *Djupedalia engeri*, which have more elongate paraoccipital processes (*Andrews, 1910*; *Brown, 1981*; *Knutsen, Druckenmiller & Hurum, 2012a*). As in *Kimmerosaurus langhami* and *Cryptoclidus eurymerus*, the paraoccipital process is expanded distally where it contacts the squamosal (*Brown, 1981*).

On the medial surface, a large anteroposteriorly oriented cavity is present at the centre of the exoccipital-opisthotic body. Although the structure is distorted, it may represent the recess for the utriculus as described for *Muraenosaurus leedsii* and *Tricleidus seeleyi* (*Andrews, 1910*). Two semicircular canal openings are visible in medial view: a dorsally orientated vertical posterior semicircular canal and a horizontal anterior semicircular canal. The posterior vertical semicircular canal is positioned just anterior to the supraoccipitial facet and runs ventrally into a cavity interpreted to be for the utriculus, similar to *Kimmerosaurus langhami* and *Cryptoclidus eurymerus* (*Andrews, 1910*; *Brown, 1981*). The horizontal anterior semicircular canal is located directly ventral to most of the prootic facet and opens posteriorly into the utricular cavity.

In lateral view (Fig. 9B), a large foramen could either be for the exit for cranial nerve X, or may be formed of multiple cranial nerve openings which have merged due to crushing.

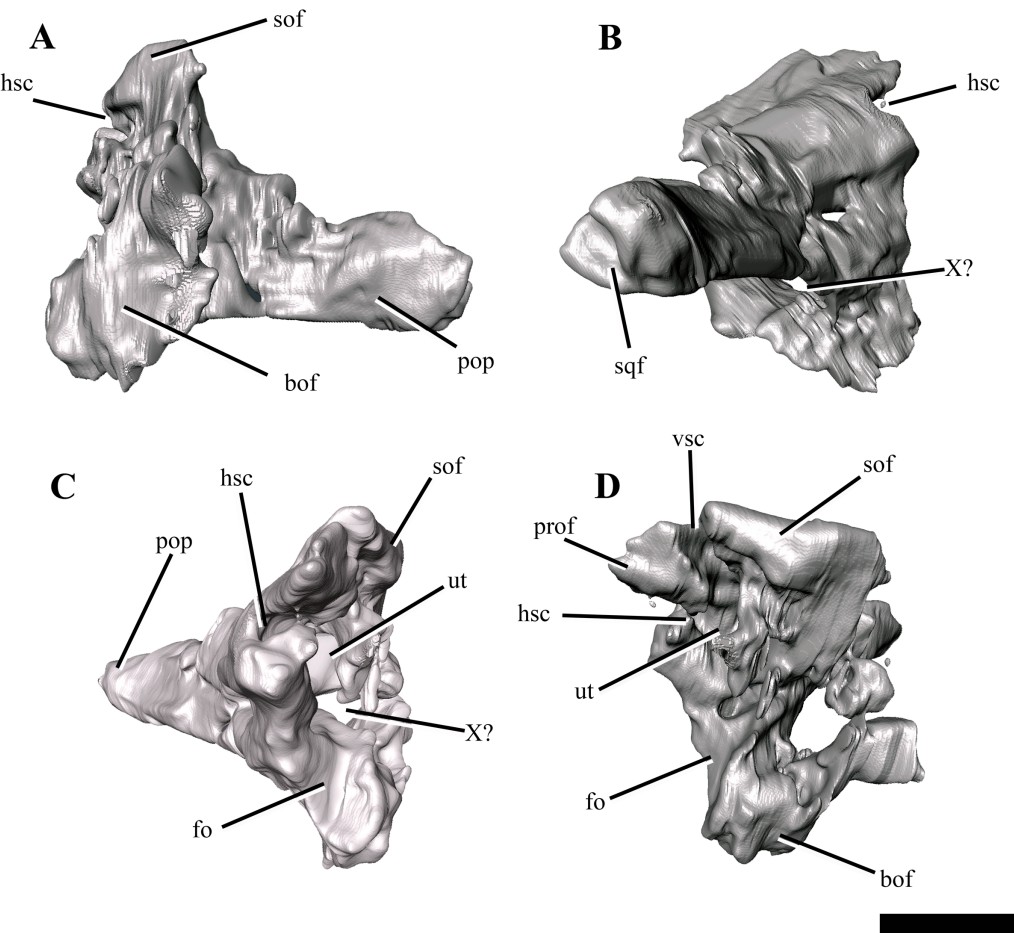

**Figure 9 The right exoccipital-opisthotic of *Ophthalmothule cryostea*, PMO 224.248 segmented out from µCT images.** In (A) posterior, (B) lateral, (C) anterior and (D) medial views. Abbreviations: bof, basioccipital facet; fo, fenestra ovalis; hsc, horizontal semicircular canal; pop, paraoccipital process; prof, prootic facet; sof, supraoccipital facet; sqf, squamosal facet; ut, utriculus; vsc, posterior vertical semicircular canal; X, cranial nerve opening. Scale bar equals ~1 cm.

The posterior portion of the fenestra ovalis is located along the anterior margin of the exoccipital, ventral to anterior horizontal canal and the prootic facet.

### Supraoccipital

The supraoccipital remains in articulation with the parietal, but has rotated anteriorly so that its posterior surface faces dorsally. The element is anteroposteriorly thickest at the exoccipital-opisthotic facet and thins dorsally. Compared to the relative dorsoventral height of the exoccipital-opisthotic, the supraoccipital contributes roughly half of the total height of the foramen magnum. The foramen magnum appears to be oval in shape, unlike the more hour-glass outline seen in *Kimmerosaurus langhami* (*Brown, Milner & Taylor, 1986*). *Ophthalmothule cryostea* lacks a posteromedian ridge on the supraoccipital, as seen in *K. langhami* and *Muraenosaurus leedsii* (*Brown, Milner & Taylor, 1986*). A small foramen is located on the midline of the dorsal border with the parietal, a feature that is

also present in *Cryptoclidus eurymerus* (*Brown, 1981*, Fig. 2), but notably absent in *M. leedsii* and possibly *K. langhami* (*Brown, 1981*; *Brown, Milner & Taylor, 1986*; A.J. Roberts, 2015, personal observations, NHMUK R.10042).

### Prootic

Two crushed and slightly disarticulated elements visible anterior to the exoccipital-opisthotics in the μCT images, are interpreted to be the prootics (Fig. 8). The elements are too distorted to warrant further description.

### Vomer

The anterior extent of the vomers in *Ophthalmothule cryostea* (PMO 224.248) is unclear, however they are mediolaterally narrowest anteriorly (Fig. 4). Posterior to the internal nares, the vomer expands in mediolateral width, becoming broadest near their posterior margin, similar to that observed in *Muraenosaurus leedsii* (*Andrews, 1910*). The left and right vomers are in full contact along the midline, though unfused and the ventral surface is convex and lacks ornamentation. This differs from the clear fusion seen in *Vinialesaurus caroli* (*Gasparini, Bardet & Iturralde-Vinent, 2002*) and partial fusion and ridged ventral surface in an undescribed juvenile Callovian cryptoclidid specimen (A.J. Roberts, 2015, personal observations, NHMUK R 2853). As in most other cryptoclidids, the vomer forms the medial and at least part of the anterior border (*Andrews, 1910*). As in some other plesiosauroids that preserve this region (e.g. *M. leedsii*; *Andrews, 1910*), the vomers have posterolaterally expanded margins that partially lie ventral to the palatines, an orientation confirmed by the μCT images in cross section. The posterior contact with the pterygoids consists of an interdigitating suture. The vomer forms the anterior border of the anterior interpterygoid vacuity, in contrast to *M. leedsii* and *Cryptoclidus eurymerus* where the pterygoids meet anteriorly along the midline and exclude the vomer from participation in margin of the anterior interpterygoid vacuity (*Andrews, 1910*; *Brown & Cruickshank, 1994*).

### Palatine

The palatine of *Ophthalmothule cryostea* forms the posterolateral border of the internal naris (Fig. 4). As is typical of cryptoclidids, the palatines do not meet anteriorly along the midline, but are separated anteriorly by the vomer (*Buchy, Frey & Salisbury, 2006*). Similar to *Cryptoclidus eurymerus* and *Muraenosaurus leedsii* a suborbital fenestra is absent (*Andrews, 1910*; *Brown & Cruickshank, 1994*). The posterior margin of the palatine is presumed to terminate at the anterior margin of the ectopterygoid, although it is difficult to discern the nature of this contact due to poor preservation in this area. The ectopterygoid area lacks a boss or flange.

### Pterygoid

The pterygoid of *Ophthalmothule cryostea* is mediolaterally narrow anteriorly and gradually increases in width posteriorly, as in *Cryptoclidus eurymerus* and *M. leedsii* (*Andrews, 1910*; *Brown, 1981*; *Brown & Cruickshank, 1994*). The anterior region of the pterygoids is separated along the midline by a prominent and mediolaterally broad

anterior interpterygoid vacuity (Fig. 4). The mediolaterally broad morphology of the anterior pterygoid vacuity is similar to *Tricleidus seeleyi* and does not narrow anteriorly to the same degree as in *Muraenosaurus leedsii* (*Andrews, 1910*).

The pterygoid forms the lateral margins of the posterior interpterygoid vacuity, which is anteroposteriorly short compared to *Cryptoclidus eurymerus* and *Muraenosaurus leedsii* (*Andrews, 1910*; *Brown, 1981*). The pterygoids do not meet along the midline posterior to the posterior interpterygoid vacuity, as is the case in some polycotylids and leptoclidids (e.g. *Edgarosaurus muddi, Umoonasaurus demoscyllus*; *Druckenmiller, 2002*; *Kear, 2006*). The posterior interpterygoid vacuity is located entirely posterior to the anterior margin of the subtemporal fossa, as in *Tricleidus seeleyi* (*Andrews, 1910*).

As in *Tricleidus seeleyi*, the pterygoid bears a narrow, prong-like anteromedial process (basisphenoid process of the pterygoid; *Andrews, 1910*) that contacts the parabasisphenoid. The anteromedial process parallels the anterolateral margin of the basioccipital, but does not form a distinct facet for it. The anteromedial process in *Ophthalmothule cryostea* is similar in relative length to that of *Tricleidus seeleyi*; however, it differs from *Tricleidus seeleyi* in being anteromedially curved rather than straight and greater in dorsoventral height (based on CT imaging) in lateral view (*Andrews, 1910*).

The quadrate ramus of the pterygoid deflects posterolaterally towards the pterygoid ramus of the quadrate and is dorsoventrally taller than wide. The pterygoid forms a broad medially facing facet for the quadrate, similar to *Kimmerosaurus langhami* (*Brown, 1981*).

### Mandible

Each mandibular ramus is disarticulated from the cranium of PMO 224.248 and the anterior portions of each are missing, including the symphyseal region (Fig. 10). Based on corresponding measurements from the upper jaws, the left mandible lacks the anterior 10.5 cm of the ramus, providing an estimated total mandibular length of 27 cm.

The left mandible, which is more complete and the basis for the following description, preserves the dentary, splenial, angular, surangular and articular (Fig. S3). The Meckalian canal is visible on the left mandible, suggesting that the prearticular and splenial are damaged. A disarticulated element either representing the prearticular or splenial is present adjacent to the right mandible (Fig. S4). There is no visible facet or suture on the surangular for the coronoid, as seen clearly in *Tricleidus seeleyi* (*Andrews, 1910*; *Brown, 1981*).

In dorsal view, the alveolar row is laterally positioned relative to the parasagittal long axis of the ramus, resulting in a mediolaterally expanded dorsal portion of the dentary that preserves fourteen alveoli (inferred from μCT). The alveoli are strongly labially angled (~60° from the parasagittal plane), which increases slightly anteriorly. This differs from the more dorsally-directed alveoli in *Tricleidus seeleyi* (*Andrews, 1910*). In dorsal view, only a couple of the primary alveoli for the replacement teeth are visible, as these are partially covered by matrix. The mediolateral expansion of the dentary preserves finished bone medial to the alveoli, contributing to at least a third of the lateromedial width of the dentary dorsal surface. Ventrally, the mediolateral expansion abruptly decreases in width. In cross section, the anterior portion of the element (at the mid-point of the dentary) is

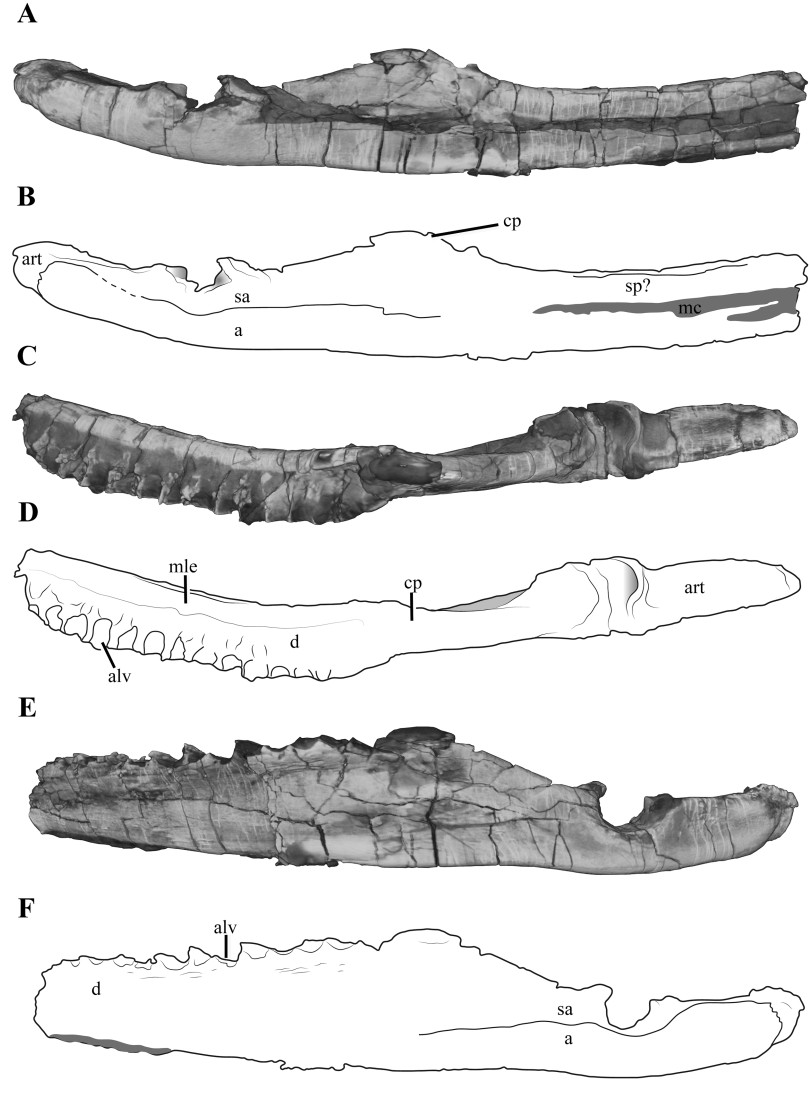

**Figure 10 The left mandible of *Ophthalmothule cryostea*, PMO 224.248 shown as μCT surface renderings and interpretations.** In (A and B) medial, (C and D) dorsal and (E and F) lateral views. Abbreviations: a, angular; alv, alveolus; art, articular; cp, coronoid process; d, dentary; mc, Meckelian canal; mle, mediolateral expansion; sa, surangular; sp, splenial. Scale bar equals 2 cm. Reconstructions and drawings by Aubrey Jane Roberts.

subtriangular due to the expanded mandibular dorsal surface. Similar expanded mediolateral dorsal surfaces are also observed in *S. larseni*, *Djupedalia engeri* and *Muraenosaurus leedsii* (*Andrews, 1910*; *Knutsen, Druckenmiller & Hurum, 2012a*, *2012b*). In *Kimmerosaurus langhami* a mediolaterally expansion of the dentary is present in one of the referred specimens (A.J. Roberts, 2015, personal observations, NHMUK R.10042), but it appears absent on the holotype specimen (*Brown, 1981*). This morphology differs from the other taxa, where a mediolateral expansion is either missing entirely (*Tricleidus seeleyi*), or a lateral expansion is only present on the posterior half of the dentary (*Cryptoclidus eurymerus*; 'Picrocleidus' beloclis) and lacks the abrupt ventral constriction

observed in *Ophthalmothule cryostea*, *Spitrasaurus larseni* and *Muraenosaurus leedsii* (*Andrews, 1910*; *Brown, 1981*; *Knutsen, Druckenmiller & Hurum, 2012b*). This feature is proposed as a new phylogenetic character (see "Discussion").

The lateral surface of the dentary is gently striated. Dorsomedially, a partial suture between the splenial and dentary is visible. Posteriorly, there is no clear suture between the dentary and surangular. The angular-surangular suture is partly visible in lateral and medial views. In *Kimmerosaurus langhami* and *Cryptoclidus eurymerus* the ventral margin of the angular is convex ventral to the glenoid, becoming concave anteriorly along the ventral margin (*Brown, 1981*; *Brown & Cruickshank, 1994*). This morphology is reduced in PMO 224.248 and *Tricleidus seeleyi*, where the ventral margin of the angular is almost straight, with a slight convexity in line with the articular (*Andrews, 1910*; *Brown, 1981*). The surface of the glenoid is slightly undulated posteriorly and dorsoventrally deep, being over half the dorsal-ventral height of the mandible. This is distinct from the shallow and smooth articular facet of *Colymbosaurus* spp., *Cryptoclidus eurymerus*, *Muraenosaurus leedsii* and *Kimmerosaurus langhami* (*Brown, 1981*; *Roberts et al., 2017*).

In *Ophthalmothule cryostea*, the retroarticular process is uniform in dorsal-ventral thickness until it reaches the posterior terminus. Similar to *Kimmerosaurus langhami* (*Brown, 1981*), the retroarticular process is nearly twice as long as it is dorsoventrally tall, in contrast to the even longer than tall retroarticular process in *Spitrasaurus larseni* (*Knutsen, Druckenmiller & Hurum, 2012b*) and *Muraenosaurus leedsii* (*Andrews, 1910*). The process is dorsally inclined at ~15° with respect to the longitudinal axis of the mandibular ramus. This is significantly less than the strong inclination seen in *S. larseni* (35°) and greater than *Colymbosaurus* indet (OUM J. 3300; ~9°) and *Tricleidus seeleyi* (10°; *Brown, 1981*). A mediolateral deflection of the retroarticular process is absent.

### Dentition

Eight partial to complete but displaced teeth, along with several fragments were found adjacent to the anterior region of the skeleton. Fully erupted teeth are absent in all of the dentigerous portions preserved in PMO 224.248, but several unerupted replacement teeth are visible in situ on the μCT images. The individual teeth vary slightly in size, but not morphology, indicating that the minor size difference represents stages of tooth replacement and not anisodonty as suggested by the alveoli. The crowns are gracile in comparison to the more robust teeth in *Tricleidus seeleyi* and *Cryptoclidus eurymerus* (*Brown, 1981*). The largest and most complete tooth preserved (Fig. 11A), measures ~4 cm in length from apex of the crown to the root. In axial (mesial/distal) view, the complete tooth (Fig. 11A), is lingually curved along the crown, straightens at the start of the root and terminates in a slightly lingually curved root terminus. This morphology differs from the significantly lingually curved teeth of *Kimmerosaurus langhami* (75°; *Brown, 1981*) and *Spitrasaurus larseni* (*Knutsen, Druckenmiller & Hurum, 2012b*). The enamelled crown represents a third of the total length of the tooth and displays a gradual transition to the root. A smaller, but fractured tooth (Figs. 11B–11E), bears fine longitudinal ridges on the enamel, which gradually fades towards the apex. The ridging is most prominent on the labial side, unlike the prominently lingually ridged teeth of *Muraenosaurus leedsii* and

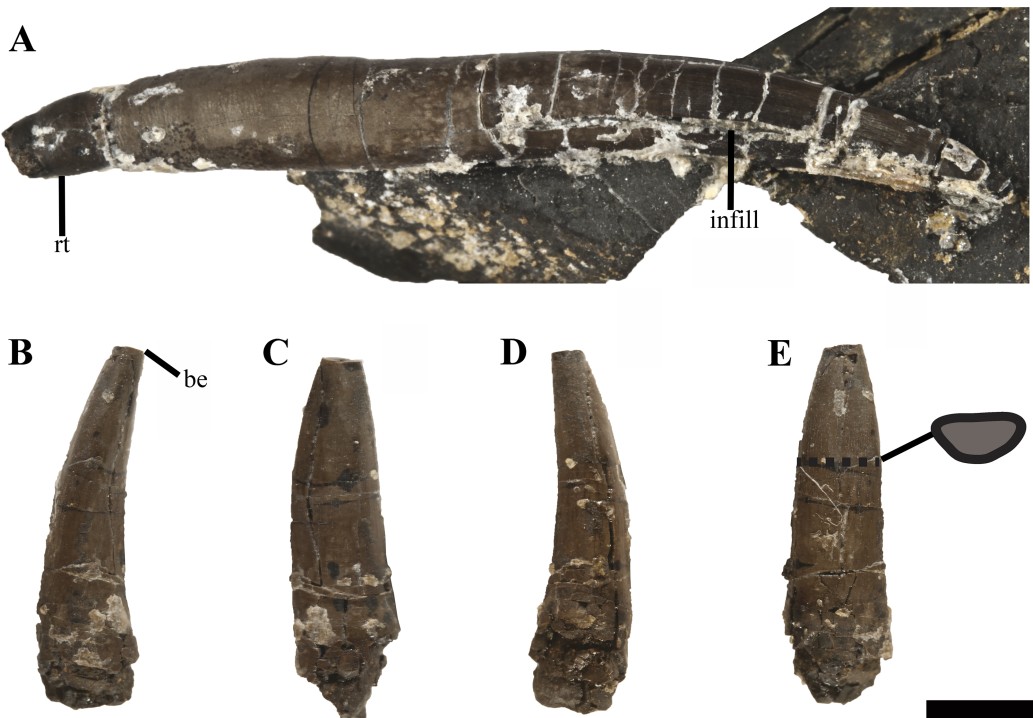

**Figure 11 Isolated teeth of *Ophthalmothule cryostea*, PMO 224.248.** (A) The most complete tooth in axial view; (B–E) an incomplete tooth in (B and D) axial, (C), lingual and (E), labial views with a cross section of the tooth. Abbreviations: be, broken edge; rt, root tip. Scale bar equals 0.5 cm. Photography by Aubrey Jane Roberts.

*Cryptoclidus eurymerus* (*Brown, 1981*). On some teeth, the mesial or distal margin of the crown bears a more pronounced enamelled ridge. This ridge could represent the edge of a partial wear facet, as it has no distinct shared morphology between teeth and has a variable presence on the preserved tooth crowns (Fig. S5). Close to the tip of the crown, the labial side is flattened compared to the convex lingual surface resulting in a D-shaped cross section, similar to *K. langhami, S. larseni* and some elasmosaurids (*Brown, 1981*; *Knutsen, Druckenmiller & Hurum, 2012b*; *Sato, Hasegawa & Manabe, 2006*). This morphology differs from the more oval-shaped cross section in *M. leedsii* and *Cryptoclidus eurymerus* (*Brown, 1981*). The shaft of the root is subcircular in cross section and slightly expanded in diameter at the start of the crown, by a gently undulating surface that decreases in width towards the root. One of the teeth (Fig. S5), has a clear reabsorption facet on the lingual side of the root. The root is straight and terminates abruptly whereas other cryptoclidids show a more gradual reduction in diameter at the root (e.g. *K. langhami* and *Cryptoclidus eurymerus*; PETMG R.283.412).

## Axial skeleton

Fifty cervical vertebrae are preserved in PMO 224.248, including the atlas-axis complex. This is greater than in Callovian cryptoclidids (*Cryptoclidus eurymerus*, 32: *Brown, 1981*; *Muraenosaurus leedsii*, 44: *Brown, 1981*) and some Tithonian—Early Cretaceous taxa (*Colymbosaurus megadeirus*, 41: *Benson & Bowdler, 2014*; *Abyssosaurus nataliae*,

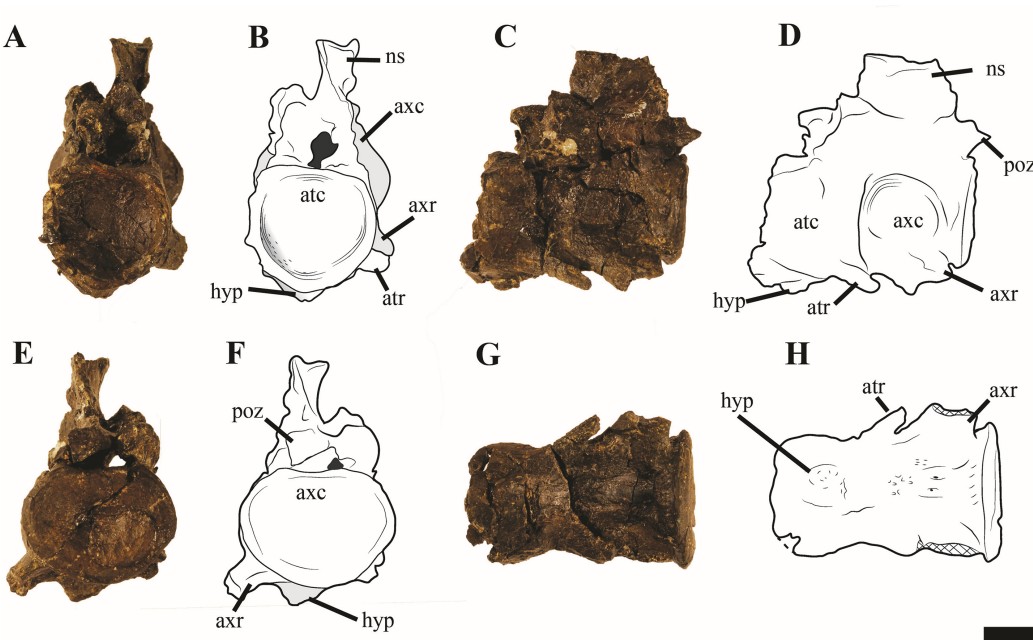

**Figure 12 Photos and interpretations of the atlas-axis complex of *Ophthalmothule cryostea*, PMO 224.268.** In (A and B) anterior, (C and D) lateral, (E and F) posterior and (G and H), ventral views. Abbreviations: atc, atlas centrum; atr, atlantal rib; axc, axial centrum; axr, axial rib; hyp, hypophyseal ridge; ns, neural spine; poz, postzygapophysis. Scale bar equals 1 cm. Photography and drawings by Aubrey Jane Roberts.           

44–51: *Berezin, 2011*; *Djupedalia engeri*, >40: *Knutsen, Druckenmiller & Hurum, 2012a*). PMO 224.248 preserves fewer cervical vertebrae than that described in *Spitrasaurus wensaasi* (60; *Knutsen, Druckenmiller & Hurum, 2012b*). In total, the neck of PMO 224.248 is estimated to be approximately 2 m in length including the preserved intervertebral spacing prior to preparation. Selected measurements from the axial skeleton can be found in the Supplemental Information (Table S2).

### Atlas-axis

Reflecting the advanced ontogenetic status of PMO 224.248, the atlas-axis complex is completely fused; however, part of the suture between the atlas and axis centrum remains visible (Fig. 12). The complex is approximately twice as anteroposteriorly long as mediolaterally wide, whereas in *Spitrasaurus larseni* and *Colymbosaurus megadeirus*, the complex is only slightly anteroposteriorly longer than wide (*Benson & Bowdler, 2014*; *Knutsen, Druckenmiller & Hurum, 2012b*). The long and narrow morphology of the atlas-axis in *Ophthalmothule cryostea* is more similar to that of *Muraenosaurus leedsii* and some elasmosaurids, such as *Aristonectes parvidens* (*Andrews, 1910*; *Brown, 1981*; *Gasparini et al., 2003*).

   In anterior view, the atlantal cup is concave and subcircular in outline. In the absence of a visible suture, it is not possible to confirm atlantal centrum (odontoid) participation in the ventral portion of the atlantal cup, a feature common in cryptoclidids, including *Colymbosaurus megadeirus* and *Spitrasaurus* spp. (*Benson & Bowdler, 2014*; *Knutsen,*

*Druckenmiller & Hurum, 2012b*). Ventrally, the atlantal intercentrum forms a low anteroventrally directed hypophyseal eminence, similar to *Colymbosaurus megadeirus*, *Abyssosaurus nataliae* and *Spitrasaurus* spp. (*Benson & Bowdler, 2014*; *Berezin, 2011*; *Knutsen, Druckenmiller & Hurum, 2012b*), that is positioned in the anterior half of the element, although this is positioned more centrally in *Colymbosaurus megadeirus* (*Benson & Bowdler, 2014*). This morphology differs from the ventral keel formed by the ventral surface of the atlas present in the Oxford Clay Formation cryptoclidids (*Andrews, 1910*) and Cretaceous elasmosaurids (*Gasparini et al., 2003*).

As in most cryptoclidids (with the exception of *Colymbosaurus megadeirus*) an atlantal rib is present and is set posteriorly on the atlas centrum (*Benson & Bowdler, 2014*). The axial rib is single-headed and occupies most of the ventrolateral length of the axial centrum, where it is fused. This differs from *Colymbosaurus megadeirus*, where the axial rib is borne partly on the posterolateral portion of the atlantal centrum (*Benson & Bowdler, 2014*). The ventral surface of the axis is generally concave, with a rounded, low and anteroposteriorly orientated ridge running from the anterior edge of the axis. The neural arch of the atlas-axis complex is fused and bears a dorsoventrally short, but anteroposteriorly elongate neural spine.

### Cervical vertebrae (3–50)

The articular surfaces of the centra are weakly amphicoelous, although not to the degree of concavity observed in *Kimmerosaurus langhami* (*Brown, Milner & Taylor, 1986*). The anterior cervical vertebrae are mediolaterally wider than anteroposteriorly long (Table S2). This relationship shifts gradually in the mid-cervical region as the length to width ratio steadily decreases posteriorly. The posterior cervical vertebrae (~35–50) are mediolaterally wider than anteroposteriorly long, unlike the more equal dimensions seen in *Cryptoclidus eurymerus* (*Andrews, 1910*) and a partial cryptoclidid specimen from Greenland (MGUH 28378; *Smith, 2007*).

The lateral surfaces of the anterior centra are conspicuously concave, becoming more convex posteriorly in the series (Fig. 13). A structure that could represent a lateral ridge is present in some of the mid-posterior cervical vertebrae in *Ophthalmothule cryostea* and is visible in the cervical vertebrae 32–38 (Fig. 14). This should not be confused with the raised convex dorsal margin of the rib facet, which is present in most of the cervical vertebrae. This transverse ridge crosses the lateral surface of the centrum, positioned in between the neural arch pedicles and rib facet. A lateral ridge may have been present in more anterior/posterior vertebrae, but cannot be unambiguously identified due to the preservation. In *Spitrasaurus* spp., a lateral ridge is present throughout most of the cervical series, located dorsal to the cervical rib facet (*Knutsen, Druckenmiller & Hurum, 2012b*). The ventral surface of the anterior—middle cervical vertebrae bear paired foramina separated by a sharp ridge in the anterior cervicals, which disappears posteriorly in the series. The presence of a ventral ridge is shared with some cryptoclidids (e.g. *Tricleidus seeleyi*; *Andrews, 1910*), but is completely absent in *Colymbosaurus megadeirus* (*Benson & Bowdler, 2014*).

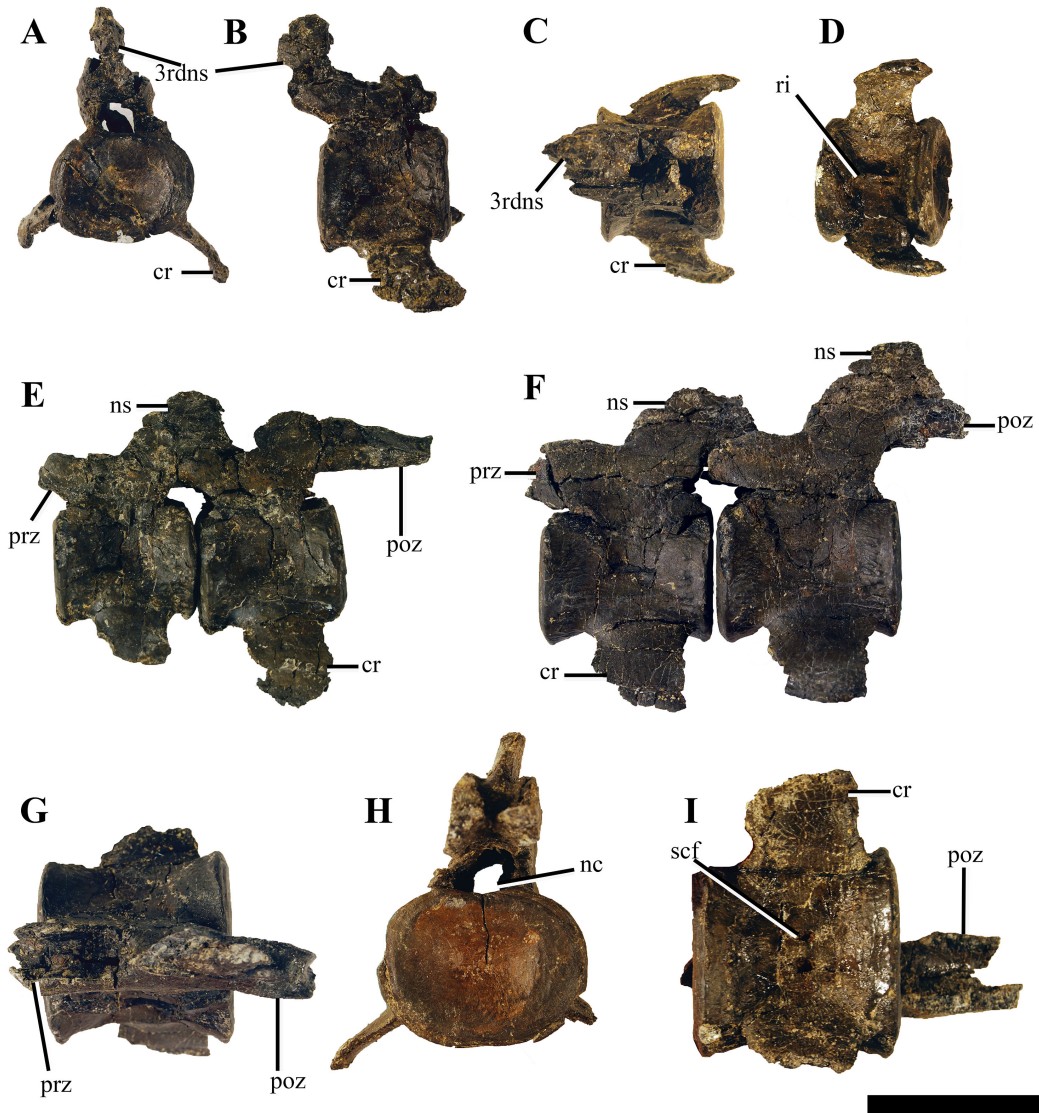

**Figure 13 Selected anterior—mid cervical vertebrae of *Ophthalmothule cryostea*, PMO 224.248.** The 4th cervical vertebrae in (A) anterior, (B) lateral, (C) dorsal and (D) ventral views, the 7th and 8 articulated cervical vertebrae in (E) lateral view, the articulated 14th and 15th cervical vertebrae in (F) lateral view, (G) the 15th cervical in dorsal view. The 17th vertebrae in (H) anterior and (I) ventral views. Abbreviations: 3rdns, neural spine from the 3rd cervical vertebrae; cr, cervical rib; nc, neural canal; ns, neural spine; poz, postzygapophyses, prz, prezygapophyses, ri, ventral ridge; scf, subcentral foramina. Scale bar equals 4 cm. Photography by Aubrey Jane Roberts.

In dorsal view, the prezygapophyses are mediolaterally narrower than the width of the centrum and positioned directly above the centrum, similar to *Colymbosaurus megadeirus*, *Abyssosaurus nataliae* and *Djupedalia engeri* (*Benson & Bowdler, 2014*; *Berezin, 2011*; *Knutsen, Druckenmiller & Hurum, 2012a*). Similar to *Kimmerosaurus langhami* (*Brown, Milner & Taylor, 1986*), in the anterior-most cervicals (3–6) of PMO 224.248, the prezygapophyses are separate along their entire length. However, in the following cervicals

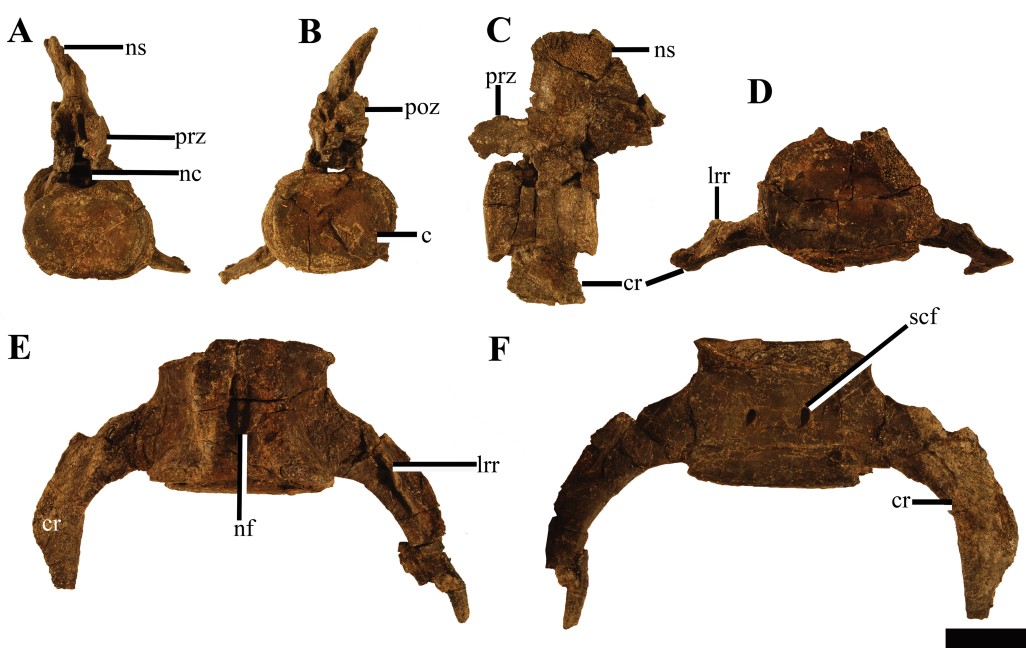

**Figure 14 Two posterior cervical vertebrae of *Ophthalmothule cryostea*, PMO 224.248.** The 29th cervical vertebra in (A) anterior, (B) posterior and (C) lateral views. The 44th cervical vertebra in (D) anterior, (E) dorsal and (F) ventral views. Abbreviations: c, centrum; cr, cervical rib; lrr, longitudinal rib ridge; nc, neural canal; nf, nutritive foramina; ns, neural spine; poz, postzygapophyses; prz, prezygapophyses; scf, subcentral foramina. Scale bar equals 4 cm. Photography by Aubrey Jane Roberts.

the prezygapophyses partially fuse medially along the anteroventral margin, and beginning approximately at cervical 15, the prezygapophyses are completely fused medially. This morphology differs from *Spitrasaurus* spp. and *D. engeri*, where the prezygapophyses are either partially or completely ventrally fused throughout the entire neck (*Knutsen, Druckenmiller & Hurum, 2012a*, *2012b*). In *Cryptoclidus eurymerus* the prezygapophyses remain unfused (*Andrews, 1910*; *Brown, 1981*). As in *Spitrasaurus* spp., the postzygapophyses are fused throughout the entire cervical series and extend posteriorly to the posterior margin of the centrum. In PMO 224.248, the length of the postzygapophyses varies throughout the series and in some regions significant posterior elongation is preserved: on the eighth cervical, the postzygapophysis length approaches the anterior-posterior length of the entire centrum (Fig. S6). When articulated with the ninth cervical, there is a larger intervertebral space in between the two centra than in preceding and following cervical vertebrae.

The neural spines in the anterior-most cervical vertebrae (3–10) are low, anteroposteriorly extended and angled posteriorly, being positioned over the postzygapophyses. Where the neural spine is completely preserved, the dorsal margin is slightly rounded. This morphology differs from the relatively straight, tall and dorsally flattened margins of the neural spines of the anterior-most cervical vertebrae in *Kimmerosaurus langhami*, *Spitrasaurus* spp. and *Djupedalia engeri* (*Brown, Milner & Taylor, 1986*; *Knutsen, Druckenmiller & Hurum, 2012a*, *2012b*). In the 7th cervical, the

neural spine is less than half the dorsoventral height of the centrum, when measured from the top of the postzygapophyses. The anterior—mid cervical neural spines (10–18) are posteriorly shifted so that the middle of the dorsal margin of neural spine is positioned directly over the posterior margin of the centrum (Fig. 13F). In lateral view the neural spines are triangular to trapezoid in outline, becoming more rectangular posteriorly and increase in dorsal-ventral height. The neural spines on the posterior cervicals are anteroposteriorly long, dorsally flattened and more centred over the centrum. Although positioned more centrally, the posterior margin of the neural spine still reaches the anterior half of the next centrum due to the anteroposterior length of the neural spine (Fig. 14). Some of the middle and the posterior cervicals show a mild anterior inclination of the neural spine. This morphology is comparable to that seen in *Spitrasaurus wensaasi* and '*Picrocleidus' beloclis* (A.J. Roberts, 2015, personal observations, NHMUK 3698; NHMUK 1965); however, the neural spines of *Ophthalmothule cryostea* do not consistently angle anteriorly as in *Spitrasaurus* spp. (*Knutsen, Druckenmiller & Hurum, 2012b*). The neural canal is oval in anterior view.

### Cervical ribs

The cervical rib facets receive single-headed ribs, which are fused to the centrum throughout the entire series (Figs. 13 and 14). In the anterior cervicals, the cervical ribs are relatively short, hatchet-shaped due to a small anterior process and terminate laterally in a posterodistal point. In the mid-cervical vertebrae, the anterior process is further reduced, gradually increasing in prominence in the posterior cervicals. This differs from *Djupedalia engeri*, where the anterior process on the cervical ribs is clearly present in the entire cervical series (*Knutsen, Druckenmiller & Hurum, 2012a*). In the mid-cervicals, the ribs are distally short and lack anteroposterior curvature. From the 40th cervical, the ribs start to elongate exceeding the length of the centrum. Similar to *D. engeri*, the posterior cervical ribs become anteroposteriorly narrower and curve posteriorly (*Knutsen, Druckenmiller & Hurum, 2012a*). This morphology differs from MGUH 28378 (Cryptoclididae indet.), where the posterior cervical ribs are straight and to *Spitrasaurus wensaasi*, where they are only slightly posteriorly curved (*Knutsen, Druckenmiller & Hurum, 2012b; Smith, 2007*). From the 44th cervical and posteriorly, the cervical ribs bear a longitudinal, dorsally positioned ridge, starting from the proximal head and terminating around the midpoint of the rib shaft, an autapomorphy of this taxon. This longitudinal ridge is also present on the pectoral and anterior dorsal ribs and likely represents a muscle attachment site (*Noè, Taylor & Gómez-Pérez, 2017*).

### Pectoral vertebrae

At least two pectoral vertebrae (Figs. 15A–15F) can be unambiguously identified (sensu *Sachs, Kear & Everhart, 2013*), with the possibility of a third (Fig. 15G). As in *Colymbosaurus megadeirus* (*Benson & Bowdler, 2014*), the centra are significantly mediolaterally wider than dorsoventrally tall in anterior view (Table S2), although this may be partially due to taphonomic compression. As in *Cryptoclidus eurymerus*, pectoral vertebrae 1 and 2 have clear circular rib facets and the subcentral foramina are widely

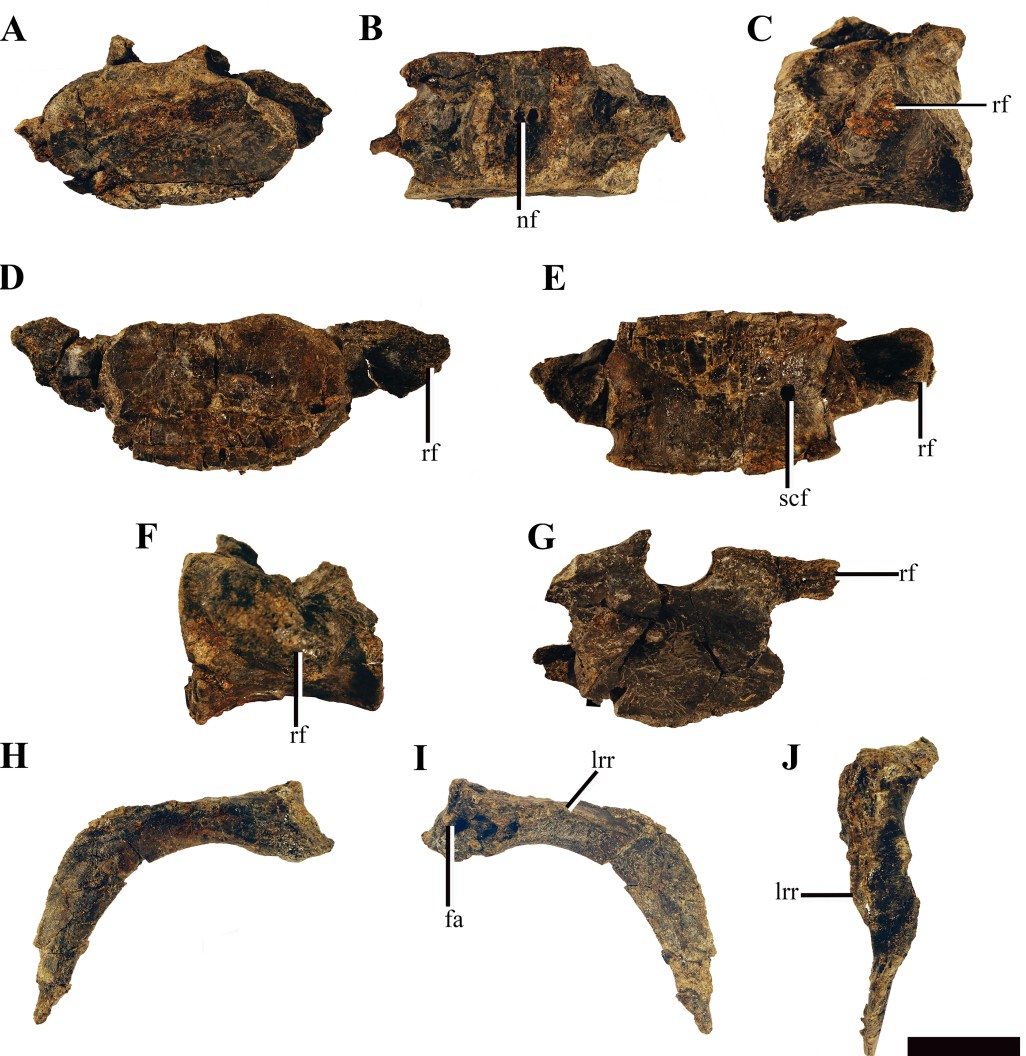

**Figure 15 Pectoral vertebrae and ribs of *Ophthalmothule cryostea*, PMO 224.248.** The 1st pectoral in (A) anterior, (B) dorsal and (C) lateral views. The 2nd pectoral in (D) anterior, (E) ventral and (F) lateral views, (G) the 3rd pectoral vertebrae in posterior view. The 1st pectoral rib in (H) anterior, (I) posterior and (J) dorsal views. Abbreviations: fa, facet for the pectoral rib; lrr, longitudinal rib ridge; nf, nutritive foramina; rf, rib facet; scf, subcentral foramen. Scale equals 4 cm. Photography by Aubrey Jane Roberts.

spaced compared to the cervicals (*Brown, 1981*). The neural arch is poorly preserved and cannot be described in detail. A third poorly preserved vertebra could represent the third pectoral (Fig. 15G). This element was slightly disarticulated posteriorly from the two pectorals and located just posterior to the medial symphysis of the scapula during preparation. The neural arch and centrum contribute to the dorsoventrally tall and laterally extended rib facet, which almost forms a transverse process on the right side. This rib facet morphology is also present in the pectoral and sacral vertebrae of *Colymbosaurus megadeirus* (CAMSM J.29640; *Benson & Bowdler, 2014*). As such the

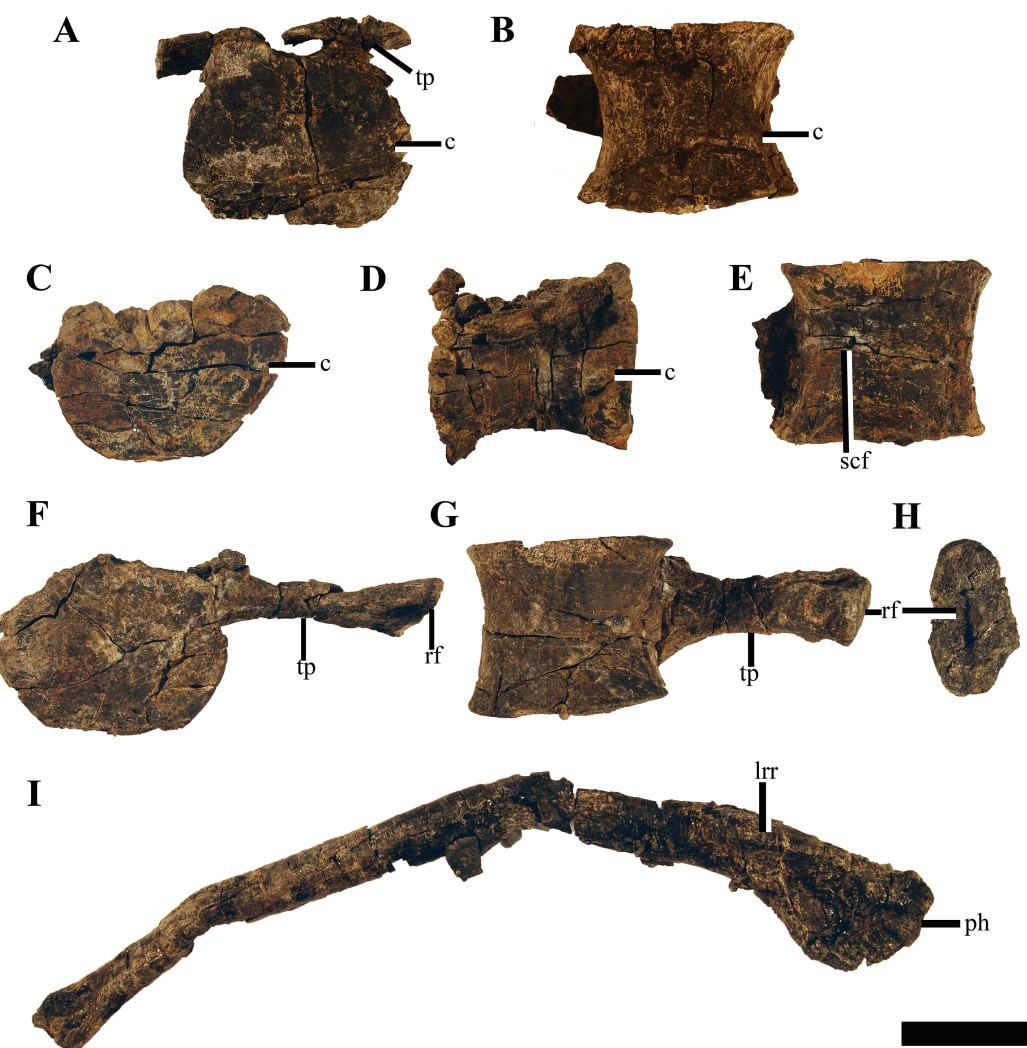

**Figure 16 Dorsal vertebrae and ribs of *Ophthalmothule cryostea*, PMO 224.248.** The 1st dorsal vertebrae in (A) anterior and (B) ventral views. The 2nd dorsal vertebrae in (C) anterior, (D) lateral and (E) ventral views. The 3rd dorsal vertebrae in (F) anterior and (G) ventral views, (H) a right anterior dorsal rib head in proximal view, (I) a complete rib in anterior view. Abbreviations: c, centrum; ph, proximal head; lrr, longitudinal rib ridge; rf, rib facet; scf, subcentral foramina; tp, transverse process. Scale equals 5 cm. Photography by Aubrey Jane Roberts.

location and similar morphology to other posterior pectorals in *Colymbosaurus* spp., supports the argument that this element can be identified as the third pectoral vertebrae.

Several pectoral ribs are preserved, either in articulation with or adjacent to the pectoral vertebrae. These share the same morphology as the posterior cervical ribs, but are more distally elongate.

### Dorsal vertebrae

Ten dorsal vertebrae are preserved, however, the posterior-most of these are poorly preserved and some are fused together by diagenesis during lithification. The dorsal vertebrae are mediolaterally narrower than the posterior-cervical and pectoral vertebrae

(Fig. 16) but not to the degree as in *Spitrasaurus wensaasi* (*Knutsen, Druckenmiller & Hurum, 2012b*). The neural arches are crushed and the transverse processes flattened; however, taking into account the shape of the dorsal rib heads and deformation of the transverse process, the rib facets are dorsoventrally taller than wide, being oval in outline as in *Spitrasaurus wensaasi* (*Knutsen, Druckenmiller & Hurum, 2012b*). This contrasts to the more circular dorsal rib facets described in *Tatenectes laramiensis* and most Oxford Clay Formation cryptoclidids (*Andrews, 1910*; *O'Keefe et al., 2011*). The anterior dorsal centra preserve either singular or paired subcentral foramina on the ventrolateral surface.

### Dorsal ribs

The majority of the dorsal ribs are incomplete due to erosion. Two ribs remain complete (an anterior (Fig. 16I) and a mid-dorsal rib), but are somewhat crushed. The rib heads are robust and single headed with an oblong ovoid facet (Fig. 16H), being dorsoventrally taller than anteroposteriorly wide as in *Colymbosaurus megadeirus* (*Benson & Bowdler, 2014*) and a specimen referable to *Muraenosaurus* (NHMUK R.2427). The mid-dorsal rib was in partial articulation with the 5th dorsal vertebrae and is 66.5 cm in actual length. This rib is curved along the proximal half of the shaft and then straightens out towards the expanded distal end. On the proximal end, a ridge is present on the posterolateral margin. The cross section is subcircular in shape along most of the shaft, but increases in mediolateral width proximally. A groove is present on the posterior surface of the proximal and distal regions of the rib. On the anterior dorsal ribs a longitudinal ridge is present, as described for the posterior cervical- and pectoral ribs.

### Gastralia

The gastral basket of PMO 214.248 is well-preserved, with at least ten sets of gastralia identified. These form a tight gastral basket, where the first set butts against the posterior margin of the coracoid. Each set contains a medial gastralium, which in turn articulates with 2–3 lateral gastralia on either side. Some of the gastralia are partially fused, which is attributed to the sideritic cement covering the dorsal surface.

### Stomach content

The posterior region of the gastral basket, was covered in a rusted silt layer in PMO 224.248. This sediment predominately silty matrix includes a large number of small worn gravel and bone fragments; thus, we interpret this area as stomach contents containing gravel. The 'gastroliths' are small, ranging from <2 cm in diameter and are significantly smaller than true gastroliths described from Late Cretaceous elasmosaurids (*Cicimurri & Everhart, 2001*; *Everhart, 2000*). A section of the layer along with a section of the underlying gastralia was μCT scanned, revealing a large amount of material embedded in the matrix. However, due to the small size of the stones, it may suggest these are simply picked up during feeding in bottom sediments (*Noè, Taylor & Gómez-Pérez, 2017*). The material requires further analysis to derive the source of the gravel and the origin of the bone material, which is beyond the scope of this article.

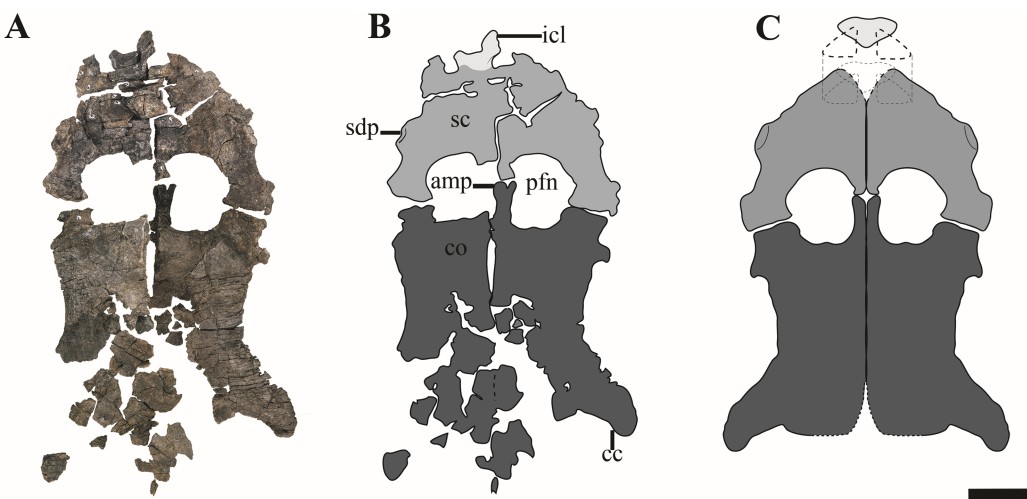

**Figure 17** **The pectoral girdle of *Ophthalmothule cryostea*, PMO 224.248.** (A) The complete pectoral girdle, (B) interpretation and (C) reconstruction. Abbreviations: amp, anteromedial process of the coracoid; cc, coracoid cornu; co, coracoid; icl, interclavicle; pfn, pectoral fenestra; sc, scapula; sdp, scapular dorsal process. Scale equals 5 cm. Photography and drawings by Aubrey Jane Roberts.

## Appendicular skeleton

### Clavicle and interclavicle

Two clavicles and an interclavicle are preserved in articulation with the scapula. Due to the hard matrix in this region, it was not possibly to separate the elements from the scapula and overlying vertebrae. The clavicles (dorsal to interclavicles) are only visible in cross section, are dorsoventrally thin and are reduced in comparison to the interclavicle, as in *Muraenosaurus leedsii* (*Andrews, 1910*; *Brown, 1981*). The interclavicle forms the anterior-most margin of the pectoral girdle along the midline, resembling the triangular element present in *M. leedsii* (*Andrews, 1910*; *Brown, 1981*).

### Scapula

The scapulae of PMO 224.248 are preserved in articulation with the rest of the pectoral elements and humeri (Fig. 17). Selected measurements of these elements can be found in Table S3. The anterior and medial margins of the scapulae are difficult to interpret, due to poor preservation and crushing by overlying elements (clavicles, interclavicle). As in all adult cryptoclidids with the exception of *Abyssosaurus nataliae*, the scapulae meet ventromedially along most of the medial margin to the posteromedial process, forming a dorsoventrally thickened symphysis (*Andrews, 1910*; *Berezin, 2011*; *O'Keefe & Street, 2009*). The posteromedial process of the scapula contacts the anteromedial process of the coracoid along an ovate facet, producing a complete pectoral bar.

*Ophthalmothule cryostea* bears a short and broad dorsal process of the scapula, in contrast to cryptoclidids from the Oxford Clay and *Plesiopterys wildi* from the Lower Toarcian of Germany, where the extension of the dorsal process can exceed half the total anteroposterior length of the element (*Andrews, 1910*; *O'Keefe, 2004*). Although the process is somewhat eroded on the right scapula, it is complete, although fractured

anteriorly on left (Fig. 17). This morphology also differs from *Abyssosaurus nataliae* and *Djupedalia engeri*, where the dorsal process forms a large part of the anterior portion of the element, being both anteroposteriorly extensive and dorsally extended (*Berezin, 2011*; *Knutsen, Druckenmiller & Hurum, 2012a*). The glenoid region is dorsoventrally thickened in comparison to the rest of the element and bears a facet for the glenoid and coracoid that are subequal in length, similar to that observed in *Abyssosaurus nataliae* (*Berezin, 2011*), but differing from that seen in *Cryptoclidus eurymerus* and *D. engeri*, where the coracoid facet is the larger of the two facets (*Andrews, 1910*; *Knutsen, Druckenmiller & Hurum, 2012a*) and to *Spitrasaurus wensaasi* where the coracoid facet is significantly smaller (*Knutsen, Druckenmiller & Hurum, 2012b*). In PMO 224.248, the glenoid facet is deeply concave, whereas the coracoid facet is nearly flat, but rugose on the articular surface.

### Coracoid

Both coracoids are articulated, although somewhat fragmented posteriorly and medially. The coracoids bear a large anteromedial process, which has a slight bifurcation anteriorly, extending further anteriorly than the scapular facet. The anteromedial process forms most of the medial margin of the ovate pectoral fenestrae, differing from *Colymbosaurus megadeirus* and *Abyssosaurus nataliae* where this margin is mainly formed by the scapula (*Benson & Bowdler, 2014*; *Berezin, 2011*) and from *Tatenectes laramiensis* where both contribute equally (*O'Keefe & Street, 2009*). The anterior portion of the medial symphysis is dorsoventrally thickened in comparison to the rest of the element, creating a shelf along the anterior margin (posterior from the pectoral fenestra), as in most derived plesiosauroids (*Benson & Bowdler, 2014*). The ventrally projecting medial symphysis of the coracoids in *Ophthalmothule cryostea* articulate along the medial symphysis so that the ventral margins form an angle close to 180°. In anterior view the dorsal margins are nearly uniform. The almost uniform dorsal surface in PMO 224.248, could be due to dorsoventral compression. This morphology is similar, although less angled than the more dorsolaterally orientated coracoids of *Cryptoclidus eurymerus* (*Andrews, 1910*; A.J. Roberts, 2015, personal observations, NHMUK R2616) and *Colymbosaurus megadeirus* (*Benson & Bowdler, 2014*; *Roberts et al., 2017*). The medial symphysis continues posteriorly throughout the preserved medial margin of the coracoid (Fig. 17). The lateral margin of the coracoid is concave and terminates posterolaterally in a distinct posteriorly curved cornu, which just exceeds the lateral margin of the glenoid in the parasagittal plane. The posterior margin is concave and a groove is present medial to the cornu, possibly to articulate with the anterior gastralia.

### Humerus

Both humeri are predominantly uncrushed and well-preserved, except for the tuberosity which is crushed on both (Table S5 for measurements). In dorsal view, the proximal portion of the humerus is angled slightly anteriorly, resulting in a slightly sigmoidal shape in dorsal view similar to some leptocleidids and polycotylids (e.g. *Hampe, 2013*; *Schumacher & Martin, 2016*). Ventrally, a prominent rugosity is present near the proximal end, forming the dorsoventrally thickest point of the humerus (Fig. 18). As in

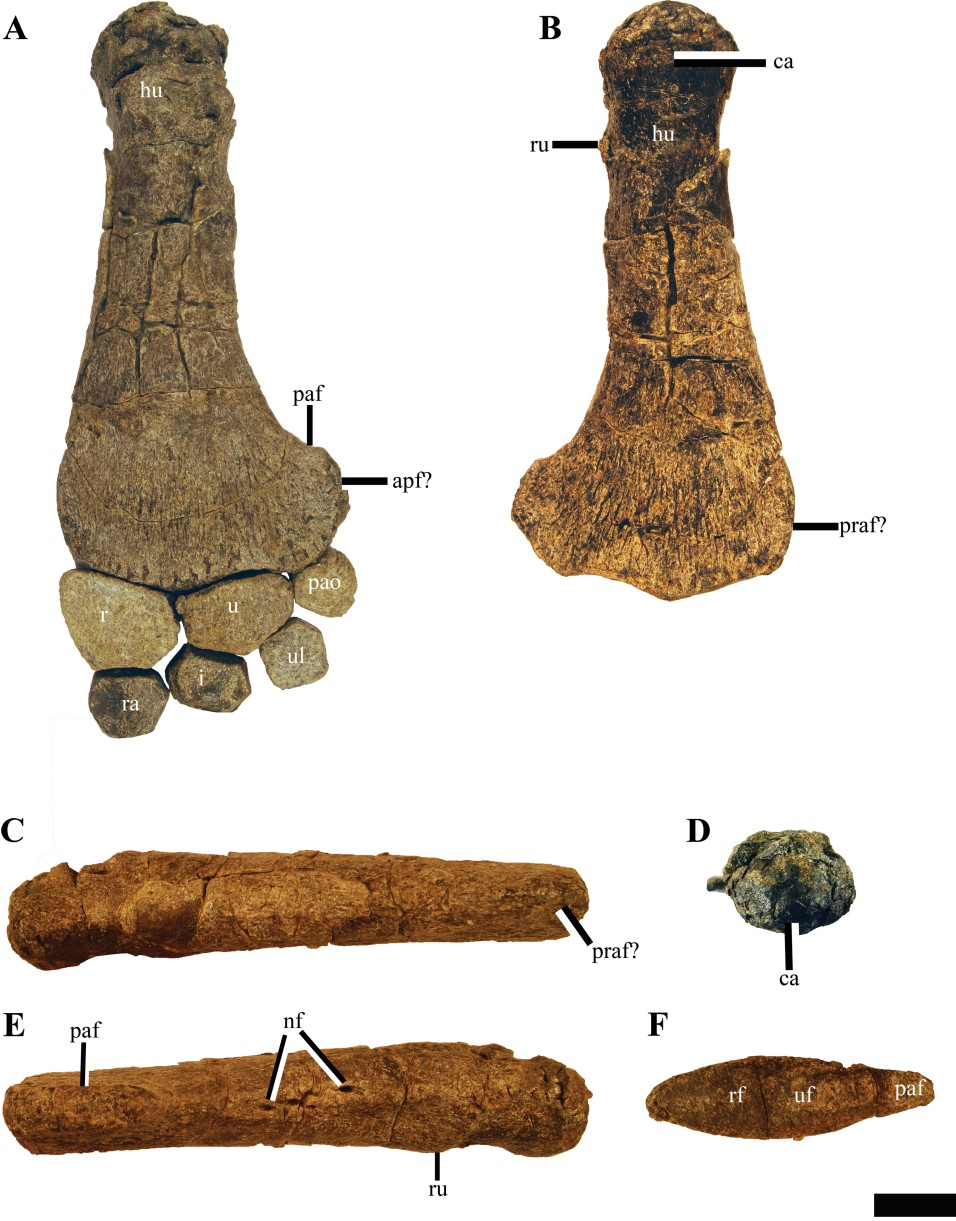

**Figure 18 The left humerus and proximal limb elements of *Ophthalmothule cryostea*, PMO 224.248.** (A) The left forelimb with the proximal elements articulated in dorsal view. The left humerus in (B) ventral, (C) anterior, (D) proximal, (E) posterior and (F) distal views. Abbreviations; apf, additional postaxial facet; ca, capitilum; hu, humerus; i, intermedium; nf, nutritive foramina, paf, postaxial flange; pao, postaxial ossicle; praf, preaxial facet; r, radius; ra, radiale; rf, radius facet; ru, rugosity; u, ulna; uf, ulna facet; ul, ulnare. Scale bar equals 5 cm. Photography by Aubrey Jane Roberts.

*Djupedalia engeri*, the anteroposterior shortest point is just proximal to the ventral rugosity, after which the humeral shaft gradually distally expands in anteroposterior width (*Knutsen, Druckenmiller & Hurum, 2012a*). This morphology differs from the more distally constricted morphology observed in *Spitrasaurus* spp., *Muraenosaurus leedsii*, *Tricleidus seeleyi*, *Pantosaurus striatus* and *Cryptoclidus eurymerus*, where the shaft is

anteroposteriorly shortest towards the midshaft and anteroposteriorly expanded at the proximal and distal ends (*Andrews, 1910*; *Benson & Bowdler, 2014*; *Brown, 1981*; *Knutsen, Druckenmiller & Hurum, 2012b*; *O'Keefe & Wahl, 2003*). Posteriorly, there are at least three nutritive foramina perforating the posterior surface near the mid-point of the shaft, an uncommon trait in cryptoclidids, but it is also observed in *Spitrasaurus wensaasi* (*Knutsen, Druckenmiller & Hurum, 2012b*).

Distally, there is little to no preaxial expansion. A large, posteriorly expanded postaxial flange is present, although not to the same degree as seen in *Colymbosaurus svalbardensis* (*Knutsen, Druckenmiller & Hurum, 2012c*; *Roberts et al., 2017*). PMO 214.248 lacks an anteroposteriorly oriented bisecting ridge on the distal epipodial facets, as observed in some specimens of *Colymbosaurus megadeirus* (*Benson & Bowdler, 2014*). The distal articular end of the humerus bears three conspicuous convex facets for the radius, ulna and a postaxial accessory element. Along the anterior margin, a rugosity is present, possibly representing a facet for a preaxial accessory element found in articulation on one of the limbs (Fig. S7). The postaxial flange has at least one facet angled posterodistally, although a secondary postaxial facet could be present directly posteriorly. Whether this posterior-most facet is an actual facet or for connective tissue attachment is equivocal, as no element was found in articulation. The distal facet morphology in *Ophthalmothule cryostea* differs from the two distal facets seen in Lower Jurassic plesiosauroids, *Microcleidus* spp. and *Plesiopterys wildi* (*Bardet, Godefroit & Sciau, 1999*; *O'Keefe, 2004*) and the Middle Jurassic taxon *M. leedsii* and (*Andrews, 1910*) the three seen in *Colymbosaurus* spp. (a single postaxial ossicle facet; *Roberts et al., 2017*) and the four suggested in 'Plesiosaurus' *manselii* (two postaxial facets; *Hulke, 1870*) and *Tricleidus seeleyi* (*Andrews, 1910*).

### Epipodials and mesopodials

The radius, ulna and postaxial ossicle element are fused through taphonomic processes to one another and likewise the radius is partially fused to the humerus. In addition, the right forelimb preserves an in situ oval preaxial element found adjacent to the preaxial facet (Fig. S8). Arguably, this element could have drifted into its current position, but based on its shape, size and position could also represent a preaxial element, and is similar to those of *Spitrasaurus wensaasi* (*Knutsen, Druckenmiller & Hurum, 2012b*). In proximal view, there is a shallow groove present on the radius and ulna for articulation with the convex distal margin of the humerus. In contrast to the Oxford Clay Formation cryptoclidids with advanced adult ossification (e.g. *Cryptoclidus eurymerus*; *Andrews, 1910*; *Brown, 1981*), an epipodial foramen (*spatium interosseum*) is absent.

The radius is the largest of the epipodial elements, being slightly anteroposteriorly wider and proximodistally longer than the ulna (Fig. 18A). This differs from *Colymbosaurus svalbardensis*, where the radius is proximodistally longer, but anteroposteriorly shorter than the ulna (*Roberts et al., 2017*) and *Spitrasaurus* spp., *Djupedalia engeri* and *Pantosaurus striatus*, where the radius is at least twice the size of the ulna in all dimensions (*Knutsen, Druckenmiller & Hurum, 2012a*, *2012b*; *O'Keefe & Wahl, 2003*). In dorsal view, the outline of the radius has a convex anterior margin which slopes posterodistally,
resembling that of *Spitrasaurus larseni* (*Knutsen, Druckenmiller & Hurum, 2012b*). In proximal view the radius is dorsoventrally thickest posteriorly and thinnest along its anterior margin. The radius has four dorsoventrally thick facets for the humerus, ulna, intermedium, radiale and an anterior facet for a preaxial ossicle (Fig. S9). The facet for a preaxial accessory element is shared between the anterior margins of the radius and radiale, as described for *S. larseni* (*Knutsen, Druckenmiller & Hurum, 2012b*). Two small elements, although disarticulated adjacent to the right forelimb along the preaxial margin, could represent part of an anterior accessory row (Fig. S8).

The ulna is anteroposteriorly wider than proximodistally long, although significantly less than the extremely anteroposteriorly elongated ulna observed in *Colymbosaurus svalbardensis* (*Roberts et al., 2017*). As in *Pantosaurus striatus* (*O'Keefe & Wahl, 2003*), the ulna of *Ophthalmothule cryostea* has five facets, with the largest facet for the humerus, two smaller anterior and posterior elements for the radius and postaxial ossicle respectively, and two distal facets of subequal size for the intermedium and ulnare.

The postaxial ossicle has three facets for the humerus, ulna and ulnare, and is convex along its posterior margin. As the postaxial element was fused to the ulna, it is possible to reconstruct its position relative to the humerus accurately. Based on this interpretation, the preserved postaxial element, occupies only a small portion of the postaxial margin of the humerus. This is somewhat different from '*Plesiosaurus*' *manselii*, according to the reconstruction by *Hulke (1870)*, where the postaxial elements occupy the entire distal postaxial margin.

The mesopodial elements were partially articulated and identified either by their position relative to the epipodial elements or their morphology. In both forelimbs all the mesopodial elements are preserved. The radiale is the largest of the three and bears five facets; the largest being for the radius, the smallest for the intermedium, two facets for the 1st and 2nd distal carpal and a facet along the anteroproximal margin for a preaxial ossicle. The intermedium bears six facets, two proximal facets for the radius and a longer facet the ulna and three smaller facets for the ulnare, 3rd distal carpal and radiale. The ulnare is bears five facets, two proximal subequal facets for the ulna and postaxial ossicle and three subequal facets for the intermedium, 3rd carpal and 2nd post axial element.

### Metacarpals and phalanges

The metacarpals were disarticulated; the distal carpals are small and their articulation to the rest of the limb uncertain. Two of the carpals are proximodistally reduced and rounded in dorsal outline. This morphology differs from the more elongated and angular distal carpals seen in most cryptoclidids (*Colymbosaurus svalbardensis*, *Cryptoclidus eurymerus*, *Muraenosaurus leedsii*, *Tricleidus seeleyi*; *Andrews, 1910*; *Brown, 1981*; *Roberts et al., 2017*). A possible 5th metacarpal was identified based on the unusual morphology of the element and on its proximal position and articulating elements. This element seems to be nearly entirely shifted into the distal carpal row.

Twenty-nine phalanges and/or metacarpals are preserved in the right forelimb and twenty-two in the left. Many of these were removed during excavation, although their

location was noted. The proximal phalanges are hourglass-shaped, with flat articular surfaces, whereas the more distal phalanges, are proximodistally shorter and more compact, similar to that observed in *Colymbosaurus svalbardensis* (*Roberts et al., 2017*).

### Femora

Fragments of the femora from PMO 224.248, were located downslope from the skeleton. These consist of fragments of the distal, mid-shaft and proximal sections of the femur and were identified based on the amount of weathering. The left limb was partly preserved with the rest of the body and was therefore more proximal to the rest of the skeleton and less weathered. The partial femur interpreted as the left, consists of a distal end, shaft fragments and part of the proximal end (Fig. S10). The bone texture and shape suggests that the femur had a postaxial flange, although not preserved. On the distal fragment of the left femur, part of the distal surface is preserved, which is smooth and slightly convex. When comparing the femoral cross-sections to the complete humeri in PMO 224.248, it is clear that the femora have a smaller circumference than the humeri along the shaft.

### Hind limb elements

Distal elements from the left hind limb, including the meso- and metatarsals and several phalanges, are preserved in PMO 224.248 and are partially articulated although heavily weathered (Fig. S10). Five mesopodial elements are preserved in left hind limb, representing the fibulare, astragalus, tibiale and the two distal tarsal elements. The 5th metatarsal appears to be entirely shifted into the distal tarsal row. Several complete and partial phalanges are preserved. As seen in *Colymbosaurus* spp., the largest element in dorsal view of the mesopodial elements is the fibulare (*Knutsen, Druckenmiller & Hurum, 2012c*; *Roberts et al., 2017*). The fibulare has six facets, with the largest being for the fibula. Along the posterior margin of the fibula there are two facets, one proximally for the postaxial ossicle and another distally possibly for a second ossicle. The astragalus is oval in outline, but bears a proximal convexity, to separate the facets for the fibula and tibia. The element is dorsoventrally thicker than proximodistally long (excluding the proximal surface convexity). The tibiale is the smallest of the three elements and bears five facets, a proximal facet for the tibia, two short proximal facets for the astragalus and second distal tarsal, a long distal facet for the first distal tarsal and a short anterior facet, possibly for a preaxial row. The distal lengths of the tibia and fibula can be estimated, based on the close articulation between the tibiale, astragalus and fibula. This suggests that at least, the distal anteroposterior extent of the fibula, appears to be longer than that of the tibia. Four metatarsals are preserved, the second, third, fourth and possibly the fifth. As in most cryptoclidids, the forth metatarsal is the largest (*Knutsen, Druckenmiller & Hurum, 2012c*).

## DISCUSSION

### Phylogenetic analysis and interrelationships of Cryptoclididae

*Ophthalmothule cryostea* (PMO 224.248) was scored into the data matrix of *Roberts et al. (2017)*, which stems from the matrix from *Benson & Druckenmiller (2014)*. Based on the results of the present study, three new characters (Characters 271–273; two cranial, one

postcranial) were created, relying on features relevant for cryptoclidids (See Data S1 for further "Discussion" and "Description"). Alternative scores and additional information on how to discern the character states of individual cryptoclidid taxa are available in the Supplemental Information. The first new cranial character relates to the presence of an interfrontal vacuity and the second relates to the dorsal surface of the dentary. We edited a previously used postcranial character related to the fibular morphology and included it in the matrix. The resulting matrix totals 273 unordered morphological characters and 76 OTUs.

The analysis was performed in TNT (V.1.5) (*Goloboff & Catalano, 2016*) using the parsimony ratchet, followed by a heuristic search using Tree bisection reconnection (TBR), that used the trees recovered from the parsimony ratchet analysis. The analysis used 1,000 iterations, with 10 random addition sequences and 10 random seed. The memory in TNT (V.1.5) was increased to hold 10,000 trees. Additional functions in TNT (V.1.5) such as *drift*, *tree fusing* and *sect. search* were not utilised. All trees were kept and auto constrain turned off and all characters were equally weighted. *Yunguisaurus* was defined as the outgroup taxon (*Cheng et al., 2006*). The *bremer* function in TNT (V.1.5) was used to calculate Bremer support (decay index). Bootstrap resampling (1,000 replications), was also performed (Fig. S5). The scripts *stats.run* was used to calculate CI and RI. The complete consensus tree for Plesiosauroidea is available in Supplemental Information (Fig. S14).

A posteriori time scaling of the strict consensus tree was comouted in the R Statistical environment (Cran) and utilised data collected from PBDB (Palaeontological Database, paleobiodb.org), in addition to personal observations from museum collections (See Data S1). The *Datephylo* function from the R package *strap* (*Bell & Lloyd, 2014*) using equal lengths was utilised to form the time scaled tree (See Data S1 for data and script).

### Results of the phylogenetic analysis

The strict consensus tree of 144 MPTs (most parsimonious trees) shows the monophyly of Cryptoclididae is relatively well supported (Fig. 19) with a Bremer support of four, as in previous studies (*Benson & Bowdler, 2014*).

Cryptoclididae is supported by the following seven synapomorphies, where only character 144 is non-homoplastic (character number/state): (144/1) the atlantal centrum participates in the anterior rim of the atlantal cup (state 0 *Tatenectes laramiensis*); 202/0 the anterolateral margin of the scapula is flat/gently convex; (235/1), in the forelimb the digits/tarsal/carpal axis extends posterodistally relative to propodial long axis because proximodistal length of radius/tibia is substantially greater than that of the ulna/fibula; (245/1), the preaxial expansion on the distal margin of the humerus is smaller than the postaxial expansion (2 in *Cryptoclidus eurymerus* and ambiguous between 1 and 2 in *Spitrasaurus sp.*); (255/3), ratio of tibia length to maximum width is >0.75 (state 1 in *Pantosaurus striatus* and state 2 in *Colymbosaurus megadeirus*); (261/2), an epipodial foramen is absent (state 0 in *M. leedsii*, 1 in 'Picrocleidus' beloclis and *Tricleidus seeleyi*, 1 and 2 in *Cryptoclidus eurymerus* depending on ontogeny, ambiguous in *Abyssosaurus*

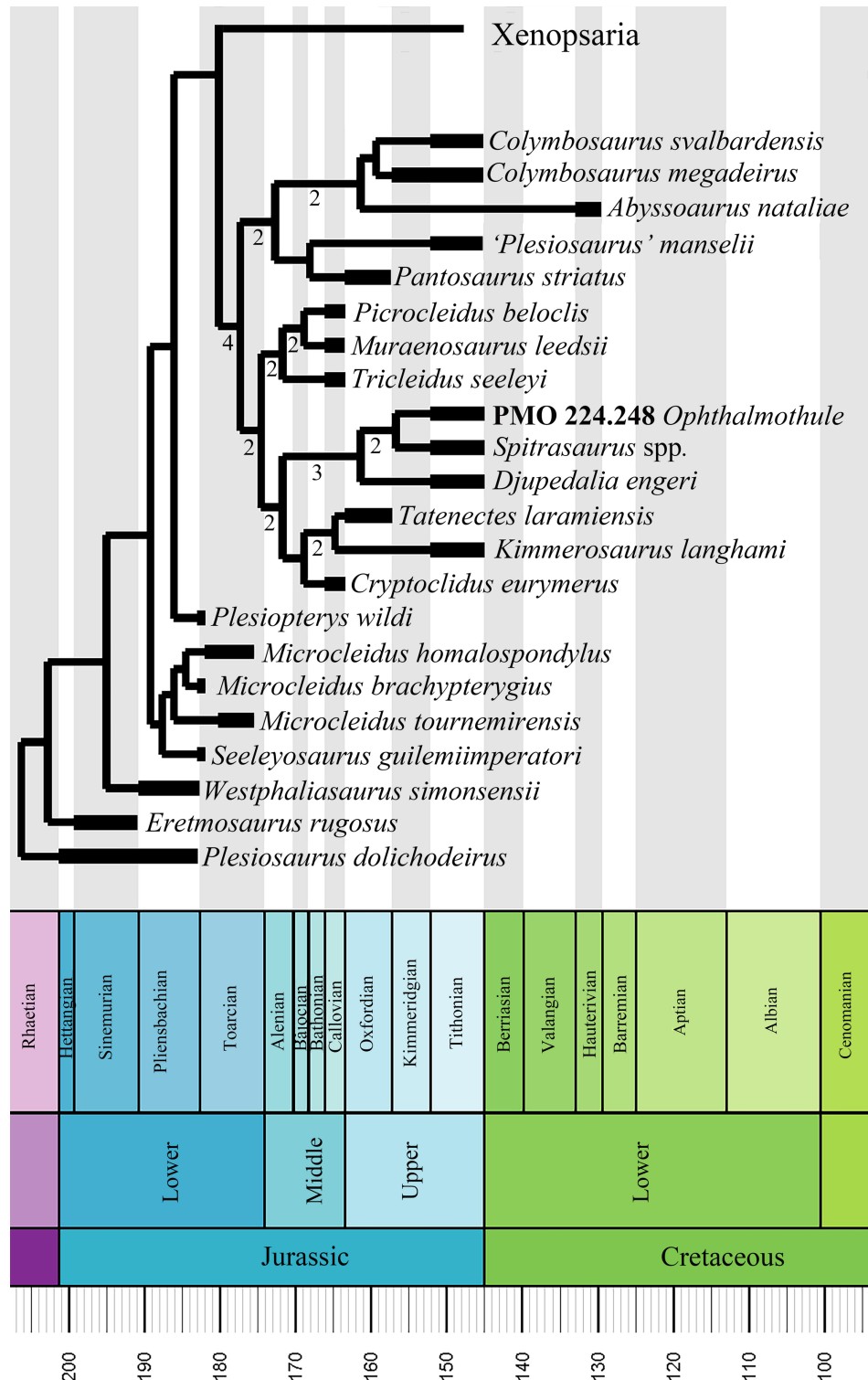

**Figure 19 Time scaled strict consensus tree of Plesiosauria with a focus on cryptoclidids.** Strict consensus tree of 144 MPTs with a tree length of 1321 (CI = 0.299 and RI = 0.663). Bremer support (>1) is shown below the clade branches in Cryptoclididae. A more complete tree including Xenopsaria is available in the Supplemental Information (Fig. S14).

*nataliae*); (262/1), the ratio of maximum radius length to maximum ulna length is between 1.4 and 1.7 (state 0 in *Ophthalmothule cryostea*, state 2 in *Cryptoclidus eurymerus*). It is noteworthy that all of these synapomorphies are postcranial features, due to the lack of complete and/or well-preserved cranial material in this clade. In the analysis of *Benson & Bowdler (2014)* only one of these characters (144) was recovered as a synapomorphy in their diagnosis. *Ophthalmothule cryostea* can be referred to the family Cryptoclididae on the basis of sharing three of seven synapomorphies (two could not be scored for *Ophthalmothule cryostea*).

As found in *Benson & Bowdler (2014)*, Cryptoclididae is split into two subclades, one of which has been formally named as Colymbosaurinae. In *Benson & Bowdler (2014)*, this subfamily includes *Colymbosaurus* spp., *Spitrasaurus* spp., *Djupedalia engeri*, *Pantosaurus striatus*, 'Plesiosaurus' manselii and *Abyssosaurus nataliae*. As cranial material for these taxa is either limited or unavailable, the synapomorphies include only post-cranial features. This is problematic, as there could be a conflicting signal between cranial and post-cranial characters in the data matrix. PMO 224.248 could be scored for a significant number of cranial and post cranial characters in the matrix. Although not recovered in the subfamily, *Ophthalmothule cryostea* shares three of the five Colymbosaurinae synapomorphies described in *Benson & Bowdler (2014)*. The addition of this new taxon to the data set, along with the three new characters, resulted in a new topology for Cryptoclididae. Although the two major subclades are still present, the Slottsmøya Member taxa *Spitrasaurus* spp., *Djupedalia engeri* and *Ophthalmothule cryostea* were recovered as a sister group to *Cryptoclidus eurymerus*, *Tatenectes laramiensis* and *Kimmerosaurus langhami*. Most of the internal is supported with higher Bremer support values than those reported in *Benson & Bowdler (2014)*, most of the internal structure of the cryptoclidid tree was not retained after running a resampling bootstrap analysis and all nodes received low support (>50). However, this analysis does show that a revision is required of the diagnostic features of Colymbosaurinae in light of the new taxon (*Ophthalmothule cryostea*). As a result, four synapomorphies for the reduced Colymbosaurinae clade were recovered, which are unique within Cryptoclididae with some exceptions: (197/0) the anteromedial margin of the coracoid does not contact the scapula; (209/2), the coracoid anteromedial process is short and subtriangular; (224/1), the anteroposterior width of the ilium is expanded, between 1.5-2.0 times the minimum anteroposterior width of the shaft and (256/1), the anterior margin of the radius is straight or convex (Also present in *D. engeri* and *Spitrasaurus* spp.).

Although the precise position of *Ophthalmothule cryostea* (PMO 224.248) as sister taxon to *Spitrasaurus* spp. is modestly supported (Bremer Support = 2), the clade incorporating *Djupedalia engeri*, *Spitrasaurus* spp. and *Ophthalmothule cryostea* is somewhat better supported (Bremer Support = 3). This clade is shares three synapomorphies: (152/5) the number of cervical vertebrae is between 50 and 60; (157/2) the anterior cervical neural spines are inflected anterodorsally (ambiguous in PMO 224.248) and (234/1), the presence of preaxial ossicles. Our results suggest that two distinct cryptoclidid lineages were present in the Boreal region during the latest Jurassic; a clade comprising *Ophthalmotule*, *Spitrasaurus*, and *Djupedalia*, and a colymbosaurine lineage

that included *Colymbosaurus svalbardensis*. However, this conclusion may change in light of new specimens from other Boreal and sub-Boreal localities.

## Palaeobiological implications

### Cranial morphology and feeding ecology

The skull of *Ophthalmothule cryostea* displays relatively large orbits compared to the temporal fenestrae (both measured as anteroposterior length; = ~1.7) in comparison with other cryptoclidid taxa for which this can be measured (*Cryptoclidus eurymerus*, ~0.93; *M. leedsii*, ~0.58). However, a comparison of orbit vs. total skull length in PMO 224.248 (=0.29), is closer, but still considerably different to that of other cryptoclidid plesiosaurs (*Cryptoclidus eurymerus*, ~0.21; *M. leedsii*, ~0.17). When compared to estimated body length (skull length/estimated body length), the skull of PMO 224.248 is considerably smaller compared to the published body size estimates in other cryptoclidids (*Brown, 1981*). In *Ophthalmothule cryostea* (PMO 224.248), the skull represents an estimated 4% of total estimated body length (5–5.5 m), while in *Muraenosaurus leedsii* specimen NHMUK R.2422, the skull is estimated to constitute 7–8% of the total body length (see Supplemental Information 3).

The mediolateral expansion of the dorsal surface of the mandible, displays an almost lateral exit angle for the teeth from the alveoli in *Ophthalmothule cryostea* and is extremely similar to the morphology seen in *Spitrasaurus larseni* (*Knutsen, Druckenmiller & Hurum, 2012b*). In *Ophthalmothule cryostea* the maximum cross-sectional diameter of one of the crowns (at crown-root transition) is ~5.5 mm and is somewhat larger to that described for *Kimmerosaurus langhami* and *S. larseni* (<5 mm; *Brown, 1981*; *Knutsen, Druckenmiller & Hurum, 2012b*). Some of the teeth of PMO 224.248 show wear facets, which could indicate a tight fit between the lower and upper jaw teeth or wear due to diet. This differs from the teeth described for *K. langhami*, where no abrasion or wear is visible on the crowns (*Brown, 1981*). The preserved teeth of *Ophthalmothule cryostea* are not as recurved as those in *S. larseni* and *K. langhami*, and therefore probably displayed a more protruding dentition than these taxa. Based on their phylogenetic position in the plesiosaurian tree, a similar morphology evolved independently in the elasmosaurid *Aristonectes parvidens* (*Gasparini et al., 2003*). However, the morphology is *A. parvidens* is more extreme as the alveoli face directly laterally in the mandible (*Gasparini et al., 2003*; *Otero, Soto-Acuña & O'keefe, 2018*).

Calculations of mechanical advantage can be used to suggest the strength of the jaw in reptiles (*Stubbs et al., 2013*; *Foffa, 2018*). Two calculations of mechanical advantage were completed on different cryptoclidid taxa (See Supplemental Information 3 for methodology and measurements), anterior mechanical advantage (AMA) and posterior mechanical advantage (PMA). The mechanical advantage of the jaw of *Ophthalmothule cryostea* was estimated to be 0.13/0.44 (AMA/PMA), using the rostral length as a proxy for the anterior extent of the mandible. It is important to mention that these calculations are computed as ratios and are thus dimensionless. This numbers are low, illustrating a low mechanical advantage, similar to *Kimmerosaurus langhami* (0.13/0.51) and *S. larseni* (0.11/0.31). In comparison, the mechanical advantage calculated was somewhat higher in

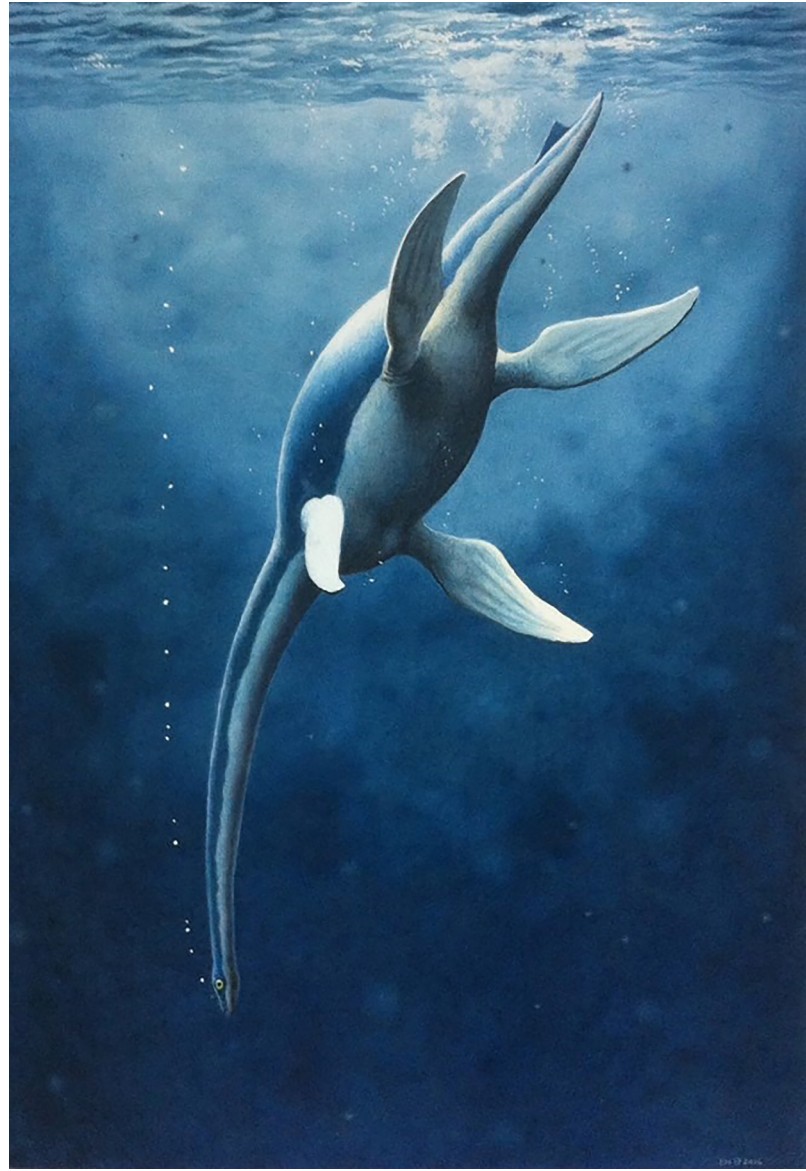

**Figure 20 A reconstruction of *Ophthalmothule cryostea* in its natural environment.** Illustration by Esther van Hulsen.                                                    

Callovian cryptoclidids: *Cryptoclidus eurymerus* (PETMG R283: 0.19/0.73), *Muraenosaurus leedsi* (NHMUK R2422: 0.20/0.64) and *Tricleidus seeleyi* (NHMUK R3539: 0.18/0.72). The mechanical advantage of these cryptoclidids is generally lower than those calculated for larger macropredatory marine reptiles (*Stubbs et al., 2013*). This suggests, particularly in the case of *Ophthalmothule cryostea* and *Spitrasaurus* spp., that these marine reptiles had a low bite force and would have been unable to eat large, armoured prey as suggested by tooth morphology.

   In vertebrates, the orbit size, which is limited by absolute constraints, cannot be reduced in size without impacting visual ability. Enlarged eyes accommodates additional photoreceptive cells and has additional area for light intake, thereby increasing the

capability of the animal to see in low-light conditions (e.g. deep, murky water, at night or during darker seasons; *Motani, Rothschild & Wahl, 1999*; *Fernández et al., 2005*; *Fischer et al., 2014*). *Ophthalmothule* was present in a northern, although not Arctic region and would have experienced seasonality (*Galloway et al., 2013*). This alone may have been the purpose for higher visual acuity, or may have been in addition to deep-diving (*Berezin, 2018*). Enlarged orbits in comparison to the temporal region are also described for the Cretaceous sub-Boreal Russian cryptoclidid *Abyssosaurus nataliae* (*Berezin, 2018*). In *A. nataliae* this morphology is suggested to be a paedomorphic feature; large orbits and short temporal region is a common juvenile feature in reptiles, including marine reptiles (*Johnson, 1977*). Phylogenetically, *A. nataliae* and *Ophthalmothule cryostea* are in two separate clades of cryptoclidids. Although this feature may not be paedomorphic, it does suggest that a high orbital/cranial ratio is present in at least two lineages of Late Jurassic–Early Cretaceous cryptoclidids.

In conclusion, to accommodate high visual acuity in the small cranium of *Ophthalmothule cryostea*, pressure to reduce the size of other areas of the cranium such as the temporal region may have been increased. However, a more likely scenario is that this odd adaptation is a result of dietary selection. In *Ophthalmothule cryostea*, this gracile and trap-like dentition in combination with the enlarged orbit relative to temporal fenestral size and low mechanical advantage, suggests that *Ophthalmothule cryostea* may have fed on small, soft-bodied prey from the water column or sea floor *McHenry, Cook & Wroe (2005)* and the large orbital size may have increased visual acuity in these environments (Fig. 20) (*Massare, 1987*; *Fischer et al., 2014*; *Noè, Taylor & Gómez-Pérez, 2017*). Further analysis into the ancestral state of skull morphology and visual acuity of this family, may be key to highlighting the changes in skull morphology in this particular lineage, but is beyond the scope of this article.

## CONCLUSION

*Ophthalmothule cryostea* (PMO 224.248) represents the temporally youngest occurrence of a plesiosaurian from the Slottsmøya Member (Agardhfjellet Formation) of central Spitsbergen. *Ophthalmothule cryostea* represents the fourth genus described from the member, although several other cryptoclidid specimens remain to be described. *Ophthalmothule cryostea* is one of the few cryptoclidids with detailed cranial osteology available, providing much needed morphological information for understanding the interrelationships of cryptoclidids. In addition, this specimen uniquely preserves a complete cervical series found in articulation, offering future possibilties to test current hypotheses on plesiosaurian neck-flexibility and evolution. As the specimen was found in the section encompassing the Jurassic–Cretaceous boundary, *Ophthalmothule cryostea* along with the Russian *Abyssosaurus nataliae* represent the youngest cryptoclidid genera in Boreal and sub-Boreal regions. The phylogenetic results of this study indicate that two separate clades of cryptoclidids were present in the latest Jurassic in the Boreal region of Spitsbergen and the sub-Boreal region of Russia.

## INSTITUTIONAL ABBREVIATIONS

| | |
|---|---|
| **CAMSM** | Cambridge Sedgewick Museum, Cambridge, United Kingdom |
| **NHMUK** | Natural History Museum, London, United Kingdom |
| **MGUH** | Geological Museum, Copenhagen, Denmark |
| **PETMG** | Peterborough Museum and Art Gallery, United Kingdom |
| **PMO** | Palaeontology Museum, Natural History Museum, Oslo, Norway |
| **OUM** | Oxford University Museum, United Kingdom |
| **SVB** | Svalbard Museum, Norway |
| **UW** | University of Wyoming, United States of America |

## ACKNOWLEDGEMENTS

The authors wish to thank the museum curators and researchers that assisted AJR during collection visits; S. Chapman and L. Steel (NHMUK), M. Riley (CAMSM), E. Howlett (OUM), M. Evans (LEICS), M. Fernández (MOZ, MLP), G. Wass (PETMG), N. Clark (GLAHM), G. Cuny (MGUH), K. Sherburn (MANCH), L.A. Vietti (UW). D. Foffa, S. Etches, V.E. Nash, D. Legg, A.S. Smith, V. Fischer, J. Wujek, E. M. Knutsen and E. Martin-Silverstone are thanked for discussion. M-L.K. Funke, C. Ekeheien, M. Koevoets and V.E. Nash are thanked for assistance during the preparation of the specimen. Ø. Hammer is thanked for assistance with the computed tomography and O. Katsamensis is thanked for access to the visualisation laboratory (μ-vis) in Southampton. M.J. Benton is thanked for comments on an earlier version of the manuscript. The reviewers V. Fischer and N. Zverkov are thanked for their comments, which helped to improve the manuscript. Gratitude is warranted to all the volunteers of the Spitsbergen Mesozoic Research Group, who participated in the 2012 excavations during their holidays. The authors warmly thank the palaeoartist Esther van Hulsen for illustrating PMO 224.248.

### Funding

This work was supported by the National Environmental Research Council and the University of Southampton graduate school. The funders had no role in study design, data collection and analysis, decision to publish, or preparation of the manuscript.

### Grant Disclosures

The following grant information was disclosed by the authors:
National Environmental Research Council and the University of Southampton Graduate School.

### Competing Interests

The authors declare that they have no competing interests.

## Author Contributions

- Aubrey Jane Roberts conceived and designed the experiments, performed the experiments, analysed the data, prepared figures and/or tables, authored or reviewed drafts of the paper, and approved the final draft.
- Patrick S. Druckenmiller conceived and designed the experiments, authored or reviewed drafts of the paper, and approved the final draft.
- Benoit Cordonnier performed the experiments, authored or reviewed drafts of the paper, and approved the final draft.
- Lene L. Delsett conceived and designed the experiments, authored or reviewed drafts of the paper, and approved the final draft.
- Jørn H. Hurum conceived and designed the experiments, authored or reviewed drafts of the paper, and approved the final draft.

## Field Study Permissions

The following information was supplied relating to field study approvals (i.e. approving body and any reference numbers):

The following permits were given by the Governor of Svalbard for the excavations in 2009, 2010, 2011 and 2012: 2006/00528-13, RIS ID 3707; RIS ID: 4760 and 2006/00528-39.

## Data Availability

The 3D data is available at Morphosource: 774.

The specimen is housed at The University of Oslo, Natural History Museum, Paleontology Museum Collections, Økern Campus: PMO 224.248.

## New Species Registration

The following information was supplied regarding the registration of a newly described species:

Publication LSID: urn:lsid:zoobank.org:pub:3578E578-4724-45FE-8CEE-C075D5C54F34.

Ophthalmothule LSID: urn:lsid:zoobank.org:act:63110850-0CAC-4DBA-99C2-7AC3B6B926DB.

Ophthalmothule cryostea LSID: urn:lsid:zoobank.org:act:97CEBF5F-58FE-472F-AFA4-9C00E37BB834.

## Supplemental Information

Supplemental information for this article can be found online at http://dx.doi.org/10.7717/peerj.8652#supplemental-information.

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
