# Peer review of "A new plesiosaurian from the Jurassic–Cretaceous transitional interval of the Slottsmøya Member (Volgian), with insights into the cranial anatomy of cryptoclidids using computed tomography"

_PeerJ, doi:10.7717/peerj.8652_

## Round 0.1 · original submission · Minor Revisions

Dear authors,

I have accepted the reviewers recommendation of ‘minor revisions’.

I look forward to receiving your revised manuscript.

·

Basic reporting

Dear editor, dear authors,

This is a great paper, describing a peculiar and very interesting new taxon. The paper meets the highest standards in all categories here.

Experimental design

I see only a couple of suggestions here:

-The comparative description is thorough and detailed but is very focussed on a a couple of well-known cryptoclidids from southern UK and Spitsbergen; while the cryptoclidid record is indeed strongly biased, I would just suggest to add some additional taxa to the comparisons, notably taxa from elsewhere as well as microcleidids and Plesiopterys, to have a sense of how derived or plesiomorphic some regions of the skeleton are in Ophthamothule. The number of comparisons also strongly diminishes when the girdles and fins are discussed.

-I have indicated in a .pdf copy of the paper instances where the wording is unclear/incorrect as well as minor questions and suggestions. I have also indicated other papers that could help with the discussion of visual acuity.

Validity of the findings

-gastroliths (p. 31). "Pebbles" are sediment grains above 20mm; under that size, you have to talk about "gravel", which is what you seemingly have here. Since it is a gravel, I think it does not works efficiently as gastroliths. The non-deliberate ingestion of sediments (diamectites?) when feeding on the sea floor appears more likely.

-This one is a rather general question/point: existence of (another) Slottmoya-only clade of marine animals. I find it puzzling that many of the distinct genera and species described by the same team cluster together very often in phylogenetic analyses. It is the case for ophthalmosaurid ichthyosaurs and is now the case for cryptoclidids as well. Don't get me wrong, it is fully possible that these environments forced a combination of (very) high cladogenesis rates and low dispersal rates. But I can't help but wondering whether this is *unconsciously* biased or a true effect. A recent reinterpretation for ichthyosaurs by Zverkov & Prilepskaya 2019 PeerJ concluded that was an effect of taxonomic oversplitting. While the distinctness of Ophthalmothule as a generic taxon appears clear, I wonder what is your take, many years afterwards and with all this new information provided by the holotype of Ophthalmothule, at the possible taxonomic status of the other cryptoclidids from Spitsbergen.

Additional comments

All the best,

Valentin Fischer

·

Basic reporting

The authors present a detailed description of a new cryptoclidid plesiosaur from the Late Jurassic (or earliest Cretaceous?) of Svalbard; the implication of µCT allows to highlight some details of otherwise hidden cranial morphology.

The manuscript is well structured and the level of English is high, however, unlike the authors, I’m not a native speaker, thus I hardly can assess this aspect of the work appropriately.

Literature is reasonably referenced. In the specific comments below, I added several additional references, some of which, in my opinion, could be beneficial for the MS.

Figures are relevant and their quality is good; they are well labeled and described. I indicated some artifacts on fig. 15 that should be cleaned. Additionally, I recommend indicating the catalogue numbers of specimens on the stratigraphic column of fig. 1; and, where possible, to be consistent with the length of scale bars - from the reader's perspective, it is hard to have scale bars of different length in every figure. It hampers an easy perception of the material.

Raw data supplied for the µCT. However, I encourage the authors to supply 3d models of processed µCT in any of the 3d formats - it will be more useful than video and screenshots provided in the current version. Additionally, I would recommend the authors to supply the dataset for their phylogenetic analysis in the nexus file format or another appropriate format for easier and faster replication of the results. Currently, there are many supplemental small word documents and jpg images. I would rather recommend making a single pdf with all these data included.

Experimental design

No comment. It is original research within scope of the journal.

Validity of the findings

I agree with the authors that the specimen represents a new species of a cryptoclidid plesiosaur. The material is good and the description is extensive and thorough. There are some minor points indicated in the marked-up pdf related to ambiguous interpretations of some observed morphologies and structures (e.g. parasphenoid, enamel ornamentation, dorsal ramus of the scapula, preaxial accessory elements) these should be clearly indicated as interpretations or illustrated more carefully so that questions regarding the validity of authors observations do not arise.

In my opinion, the most problematic issue in the current version of the MS is the speculation on the stratigraphic position of the specimen and consequently on the impact of the new specimen on the understanding of the Jurassic-Cretaceous transition in plesiosaurs. After reading the MS, I do not see how the specimen shed light on the supposed turnover. There is no discussion on the transition in plesiosaurs and other fossil groups and the research of this issue is not discussed and cited appropriately. In current form, it is not appropriate: either the expanded discussion should be added, or phrasings regarding the impact of this research for the understanding of the Jurassic–Cretaceous faunal turnover should be reduced if not completely omitted in the MS (see my comments to lines 37-40).
The stratigraphic position of the specimen is ambiguous. As with other specimens from Slottsmøya, authors use the relative position (in meters) to a “marker bed” as a definer of a stratigraphic level. However, considering the large area of excavations and that the thickness and composition of the member vary significantly even in the studied area (for details see my comments for lines 73–77), such an approach is not effective and results in a dubious stratigraphic position of all Svalbard marine reptile materials. As the authors correctly noted, in the Boreal Realm there is Volgian stage, which poorly correlates with the Tithonian international unit, except for its base. Actually, it is hardly possible to say to which part of the Tithonian the Slottsmøya Member corresponds, even having precise biostratigraphic data (authors have no biostratigraphic data at all). Authors declare that there are some data for Early Cretaceous age of seeps, located stratigraphically above the level with the specimen (with reference to another research, but no details) – this phrasing is too general and imprecise and such a data has no significant impact for understanding the age of underlying sediments. Actually, there is no robust evidence that the specimen was located on to the Jurassic–Cretaceous boundary or in the Cretaceous part of the outcrop. What can be said, is only that the specimen is closer to the J/K boundary than most other plesiosaur specimens from the locality. In this regard, the conclusion (lines 1141-1143) that “the phylogenetic results of this study indicate that two separate clades of cryptoclidids crossed the Jurassic – Cretaceous boundary in the Boreal region of Spitsbergen and the sub-Boreal region of Russia” is not well stated.

Additional comments

It is nice to see the study of this astonishing specimen on its final way to publication. My congratulations to the authors! I enjoyed the extensive and detailed description and good figures. In my opinion, this work should be accepted for publication in PeerJ after minor revision. I hope my remarks and advice will be of some help in improving the accuracy of the paper. Looking forward to seeing it published!

Below is a list of all my specific comments (for making authors work on the revision of the MS easier these could also be found in the attached marked-up pdf):

Line 1. “boundary”
I suggest replacing with “transitional interval”
J/K boundary is a problematic case and your specimen is clearly not from a boundary, but from an interval, where the boundary is presumed to be.

Lines 18-19
Not clear. How you count this? 'Taxon' is a broad term implying a taxonomic group of any rank. In such a phasing it appears that you consider a taxon of specific rank, presumably genera. When considering not only species but also supraspecific taxa - there will be more than 13... Another problem is that it is not clear that you consider valid taxa, because there are much more than 13 "known cryptoclidid taxa", but not all of them are recognized as valid currently.
Please, list the ‘taxa” in parentheses. I wonder what these five are. Even for genera, there are more than five: Cryptoclidus (good stuff), Tricleidus (good stuff), Picrocleidus, Muraenosaurus (great stuff), Tatenectes, Vinialesaurus, Kimmerosaurus (good stuff), Spitrasaurus, Djupedalia, Abyssosaurus - 10

Rephrase, please. You can use e.g. "genera" in both sentences to avoid confusions.
Also not really clear what is 'complete' for you (what percentage of completeness would be enough to be 'complete'? 100-90? >80?). For many genera of OCF cryptoclidids there are very good cranial materials (Muraenosaurus, Tricleidus, Cryptoclidus). So which cryptoclidid, in your opinion, is the only that "has a complete, but compressed cranium". Please add its name in parentheses. Your new specimen has incomplete skull as well... as some parts (postorbitals, jugals, medial squamosals) are either missing or too poorly preserved.

L 23 add “gen. et sp. nov.” after the taxon name

L 31-32
Not demonstrated in the MS. You have no robust evidence that the new plesiosaur is Early Cretaceous, but not latest Jurassic, thus still the only robust evidence for boundary crossing is Hauterivian Abyssosaurus. I would suggest deleting this highlighted part.

L 37-40
I do not think that this is an appropriate formulation. Patterns of faunal turnover during the Jurassic–Cretaceous boundary interval are contentious, and potentially varied among different groups and environments. In fact, the lack of consensus over the importance and severity of faunal turnover during this interval suggests that it does not represent a discrete mass extinction and may not have differed from background patterns of turnover.
"As with most other marine reptile groups, plesiosaur taxonomic diversity was heavily affected by eustatic sea-level changes during the Jurassic – Cretaceous transition" . You know that it is not the case for ichthyosaurs (Fischer et al., 2012; Zverkov & Prilepskaya, 2019). Recently we also discussed the turnover in pliosaurids (Zverkov et al., 2018) and demonstrated that it is complex and there was no decrease in disparity during this transition... So not really clear for me which "most other marine reptile groups" you imply here...

L 46
It seems that something is missing here... Furthermore you are citing a paper on ichthyosaurs - inappropriate citation, please replace. You can cite your previous paper (Roberts et al., 2017) and numerous papers of Espen Knutsen et al (2012)...

L 47
why brackets? I would suggest neither parentheses no brackets, but "Colymbosaurinae Benson et Bowdler, 2014" as a first mention of the taxon name with reference to authors and year as it should be when mentioning the taxon for the first time in the MS. Please also check other taxa for this.

L 55
Incorrect phrasing. 1 - you are writing "representing a new genus" but then giving a species name - so either give only a generic name or write "representing a new genus and species" 2 - optional and my subjective opinion - from the sentence it appears like the specimen is significant because it is a new taxon, but it is an outcome of its uniqueness, not a reason. It is unique because it has a number of unique features that allow you to propose a new genus and species.

L 58
Cryptoclidid genus?
Yet again a confusion with counting
You have three cryptoclidid genera and four species already in the Slottsmøya Member. And much more described specimens. So here you describe representative of the fourth genus and fifth species and ??? specimen of cryptoclidid from the member.

L 61-62
After reading the MS I don't see how the specimen shed light on the supposed turnover. As there is no discussion on the transition in plesiosaurs, it should be either added, or such phrasings should be omitted in the MS.

L 73-74
12 Myr is obviously a mistake. On what data this estimation is based? Even in the cited work (Hammer et al. 2012, fig. 7), the Slottsmøya Member spans less than 6 Myr. Actually, the whole Tithonian was 7 Myr by the results of recent research (Lena et al. 2019). Here you are speaking about "upper Tithonian" to lower Berriassian time interval - in no way it can be 12 Myr. Another problem is that, as you correctly noted, you have Volgian, which poorly correlates with the Tithonian, except for its base. Actually, you cannot say to which part of the Tithonian the Slottsmøya Member in your section corresponds, as you have no biostratigraphic data. You can say that the Slottsmøya Member corresponds to middle to upper Volgian (for the biostratigraphic correlation of the base of the member see Rogov, 2010), which, in its turn, could be approximately correlated with the interval from the middle of the lower Tithonian to the lowermost Berriassian (see e.g. Rogov 2014).

L 75. Add “On Janusfjellet (central Spitsbergen), the member is divided...”
This 'division' in Collignon & Hammer (2012: 92) is for Janusfjellet. In Wimanfjellet, from which PMO 224.248 derives, there is different section already (see Dypvik et al., 1991, fig. 5) - from that the scheme it appears that there is no such units in Wimanfjellet, or the thickness of these units is significantly different as the whole thickness of Slottsmøya on Wimanfjellet exceeds that of Janusfjellet 1,4 times! (Dypvik et al., 1991, fig. 5). This should be kept in mind (the best way, probably, is to omit the sentence). The thickness of the Slottsmøya Member varies, generally from 70 to 100 m, and is 84 m in the stratotype at Slottsmøya, western Sabine Land (Dallmann (Ed.), 1999, p. 190), thus the division on three units by meters for 52 m thick Slottsmøya Member of Janusfjellet is actually inapplicable for other outcrops of the member on the archipelago.

L77 “The specimen (PMO 224.248) described here derives from the upper unit (at 38.5 m)”
As I wrote above, that unit is found on Janusfjellet. You have no robust data for the presence of the same units and their volume on Wimanfjellet, thus the only thing you can say is that "In case if found on the Janusfjellet, PMO 24.248 would have been within the upper unit..."

L 164 “narrow sagittal crest”
mediolaterally
Here and throughout the MS, please indicate the directions and orientations when needed in order to avoid confusions.

L 167 – same as for L164

L 181 “faint longitudinal ridged teeth, distinct on labial side”
longitudinally ridged teeth, or longitudinal ridges on teeth?
not clear from your figures. It would be great to have magnified regions of the enamel on the labial and lingual surfaces to make this clear. In fact, it sounds really strange to have ridges on the labial surface but not on the lingual. To my knowledge, it is more common in plesiosaurs when the labial surface is smooth and the lingual is ridged.

L 184 “hypophyseal eminence” vs “ventral keel”
It is a terminological issue and should be made clearer. Benson & Bowdler (2014) call morphologies with the keel "hypophyseal eminence" as well, so not really clear what you mean in this context as it seems like 'ventral keel' and 'hypophyseal eminence' are different morphologies for you, but boundaries between the two states are not defined, e.g. "However, the hypophyseal eminence of Kimmerosaurus is a narrow ridge, and extends along the entire ventral length of the atlas-axis complex. This is unlike the anteriorly located, mammilate hypophyseal eminence of Colymbosaurus" (Benson, Bowdler, 2014, p. 1066)

L 196 mid-posterior cervicals are not known for Kimmerosaurus, at least in published works

L 199 “short and reduced”
I would not be so sure. It seems more like broken and lost than reduced.

L 200 Add Spitrasaurus, Djupedalia, 'Plesiosaurus' manselii,
How do you choosing which taxa to list? Throughout the diagnosis you listing 2-3 taxa with different morphologies even when there are much more of them bearing a certain trait… Please reread your diagnosis and try to consider all the know taxa when certain trait could be compared.

L 201 As you have only fragments of the femur preserved, this assumption is speculative – thus not for the diagnosis. Even considering your statement in the description that the diaphyseal cross-section of the femur is not as large as in the humerus... From my knowledge, in plesiosauroids femora commonly have more slender diaphyseal part than the humeri, so this suggestion regarding the size based on such assumption means nothing and should not be in the diagnosis. Even though it is very likely, considering other cryptoclidids, that your suggestion that the femora were smaller is correct.

L 202 “sigmoid humerus in dorsal view*”
Considered autapomorphic. However, this is too general formulation. The curvature is too slight if present (I do not see this curvature compared to that pronounced in xenopsarians). Such an outline is not unique among Cryptoclidids. Similar outline e.g. in Picrocleidus beloclis holotype
“forelimbs” - humeri - there are many distal facets in forelimbs

L205 - This section is better fitting for Materials. Not for Systematic Palaeontology

L 206 “near-complete” - I would not say that the near lack of pelvic girdle, hindlimbs, and a tail is insufficient loss and that with such a lack the skeleton is still near-complete. You can only suggest that it was near complete before being eroded...

L234 “Fig. 3.3” – correct this

L238 “rugose” - same, but even rougher in Tricleidus...

L 263 – five pmx teeth not only in Muraenosaurus, but also in Vinialesaurus caroli and Tricleidus seeleyi

L 280 “heterodonty” - anisodonty. Although these terms are commonly mixed up, it is better to use isodonty/anisodonty when speaking of size of the dentition and homodonty/heterodonty when assessing its morphology.

L 348 “tall and sharp crest” also in Tricleidus seeleyi and Muraenosaurus leedsii

L 365 “anterior” – are you sure?

L 398 “posterolateral” - maybe anterolateral?

L 426
I would not be so sure regarding the presence of additional anterior articulation having CT results of such poor quality. In your models, I do not see the morphology - only irregular fragments that are asymmetrical in addition.

L 437 “the parabasisphenoid is very thin and somewhat damaged” - Indeed. I would rather suggest that parasphenoid is detached and lost in your specimen so that the element you describe as parabasisphenoid is a body of the basisphenoid solely.

L 440 “lacks a projecting cultiform process”
Please consider the case that the parasphenoid can be too poorly preserved for correct processing or even lost (considering the irregular surface with cracks and breakages in this region) and you interpret and describe an incomplete palate as complete...

L 467
I doubt this is due to crushing - it is normal for plesiosaurs to have a common jugular canal (i.e., canal for X and XI nerves, and for the jugular vein)
From L486- (Vomer, Palatine, Pterygoid, )
add references to figures somewhere in the sections

L 492 “Iturralde-Vincent» - check the spelling – Vinent

L 561-562
“possibly due to taphonomy” - you can say the same regarding PMO 224.248. Based on my observations on this nicely preserved specimen it is more likely a natural condition than a taphonomic distortion.
“it appears absent on the holotype” - the mandible of the holotype is disarticulated. The dentaries are nearly completely preserved and fused. their posterior ends are mediolaterally wide. It is possible that this could be due to taphonomy, but I would rather suggest that this is a natural condition further supporting your observation of this morphology in the referred specimen.
Photograph of dentaries of the holotype - https://www.dropbox.com/s/g40mop6l28nuyka/IMG_5879.JPG?dl=0

L585 what about Kimmerosaurus?

L 593 anisodinty

L 603 “ridging is most prominent on the labial side”
Could you provide close-up photographs of the enamel ornamentation as supplemental files, please? This interpretation sounds confusing to me and it would be nice to have pictures for agreeing that your description is correct.

L 609-612
Incorrect interpretation of Kimmerosaurus dentition. In this taxon, the crown cross-section is markedly D-shaped and strongly labiolingually compressed. Please check the holotype, or you can cite this as my personal communication. In OCF cryptoclidids circular in cross-section indeed.
By the link are my photographs of Kimmerosaurus langhami holotype teeth:
https://www.dropbox.com/s/qdq4sut9zmg6esp/IMG_5894.JPG?dl=0
https://www.dropbox.com/s/h2yklqpxjvx0as8/IMG_5896.JPG?dl=0

L 697 This is quite a speculative suggestion. The morphology of zygapophyses in sauropod cervical vertebra and in plesiosaurs are significantly different. The morphology you describe for PMO 224.248 would be a significant constraint for flexibility.

L 715 and other mentions of this taxon – Picrocleidus in quotation marks
Although synonymized with Muraenosaurus by Brown, its validity is supported in recent studies - I see no reason to keep this generic name in quotation marks.

L 729 “Cryptoclididae indet” why in italic?

L 784 In my opinion, this section adds nothing to the description and is too superficial. It could be moved to Materials section along with Taphonomy.

L 791 see O'Keefe et al., 2009 A plesiosaur, containing...

L 816 Are you sure in this interpretation? From your figure, it seems more likely that the dorsal rami of scapulae are broken and lost, rather than reduced

L 853 “resulting in a sigmoidal shape” add somewhat... slightly... the thing is that it is not as marked as in xenopsarians.

L 873 From your figure, it seems that there is many bones around the limb so why you are so sure that this is a preaxial accessory element, but not e.g. displaced postaxial accessory element. Don't you think that it is strange that the whole distal forelimb is disarticulated and displaced, but a small ossicle with loose contact is still in situ?

L 878 -Four distal facets in Tricleidus with corresponding two postaxial accessory elements! See Andrews, 1910, fig. 77

L879 “‘Plesiosaurus’ mansellii” – manselii check spelling here and throughout the MS

L883-885 see comment for L 873 above. This is not really robust evidence for pae

L914-915
Please delete the sentence as it is speculation
(i) - there are two elements in 'P.' manselii, and one preserved in PMO 214.248 - doesn't mean there was no second or even third pae.
(ii) - the posterodistal edge of the humerus in 'P.' manselii is broken and lost, thus its morphology is unknown

L 953 – “S.11” – 10

L 945 “tibiale” and “astragalus”
if you have astragalus, you can’t have the tibiale the same time. either interpret it as a centrale, or apply a traditional interpretation with the intermedium (not astragalus) and tibiale.
I had no time for publishing this, but discussed at SATLW-2017 - https://www.researchgate.net/publication/319454783_MESOPODIAL_ELEMENTS_IN_HINDLIMBS_OF_ICHTHYOSAURS_AND_PLESIOSAURS_CONTROVERSIAL_INTERPRETATIONS_AND_POSSIBLE_SOLUTION

L 1009 – “Abyssosaurus laramiensis” - check taxon name

L 1022 - nice statement, especially after the paragraph listing 7 postcranial synapomorphies of cryptoclidids and no cranial. Everything could be supposed and called problematic when some parts are missing. Still you have no sufficient reasons for this suggestion.

L 1034 “in light of the new taxon (PMO 224.248)”
If you write 'new taxon', you should give its name in the parentheses, but not the specimen number.

L 1035 Please, mark which of the synapomorphies are non-unique

L 1047 “were present in the Boreal region during the Late Jurassic – Early Cretaceous.”
Only "during the Late Jurassic". You have no robust evidence for the presence of the second lineage in the Early Cretaceous

L 1071-1074
If you intended to mention aristonectines, please cite also the recent papers with relevant discussions of this issue O'Keefe et al., 2017; Otero et al., 2018

L 1079 mention for the clarity that the mandible is incomplete and this estimation is approximate

L 1104
It could be nice if you tried to add your new taxon to the dataset of Foffa et al., 2018 and see if it occupy the close dental morphospace to e.g. Kimmerosaurus...

L 1112
Although the subject of this section is important and extremely interesting, in its current form... this is a very short and superficial part that adds nothing to the paper. I would rather suggest to remove it, or expand so that it will be more sound. Maybe by adding some sort of analysis? Have you seen a paper by Soul & Benson (2017)?

L 1139 “and likely Berriasian in age” no idea why do you think so. It is not impossible, but you have no evidence

L 1257 not cited in the text

Comments to supplemental information

I like the new characters you added to the dataset.
In the section “The impact of the new characters” there is nothing regarding the IMPACT of character 272: Dentary, mediolateral expansion of the dorsal surface.

“Table S1.1: Table over the first and last occurrences of taxa.”
“Data from PBDB with some geological age mistakes corrected.”
Please explain, what exactly and why was corrected.

Literature cited in the review:
Benson R.B. J., Bowdler T. 2014. Anatomy of Colymbosaurus megadeirus (Reptilia, Plesiosauria) from the Kimmeridge Clay Formation of the U.K., and high diversity among Late Jurassic plesiosauroids. Journal of Vertebrate Paleontology, 34:5, 1053-1071
Dypvik, H., Nagy, J., Eikeland, T.A., Backer-Owe, K., Andresen, A., Haremo, P., Bjærke, T. & Johansen, H. 1991. The Janusfjellet Subgroup (Bathonian to Hauterivian) on central Spitsbergen: a revised lithostratigraphy. Polar Research, 9 (1): 21-44.
Dallmann W.K. (Ed.) 1999. Lithostratigraphic Lexicon of Svalbard. Upper Palaeozoic to Quaternary Bedrock. Review and Recommendations for Nomenclature Use, Committee on the Stratigraphy of Svalbard/Norsk Polarinstitutt, 320 pp.
Fischer V, Maisch MW, Naish D, Kosma R, Liston J, Joger U, Krüger FJ, Pardo Pérez J, Tainsh J, Appleby RM. 2012. New ophthalmosaurid ichthyosaurs from the European Lower Cretaceous demonstrate extensive ichthyosaur survival across the Jurassic–Cretaceous boundary. PLOS ONE 7(1):e29234
Foffa D, Young MT, Stubbs TL, Dexter KG, Brusatte SL. 2018. The long-term ecology and evolution of marine reptiles in a Jurassic seaway. Nature Ecology & Evolution 2: 1548–1555.
Hammer Ø, Collignon M, Nakrem HA. 2012. Organic carbon isotope chemostratigraphy and cyclostratigraphy in the Volgian of Svalbard. Norwegian Journal of Geology 92:103–112
Knutsen, E. M., P. S. Druckenmiller, and J. H. Hurum. 2012a. Redescription and taxonomic clarification of ‘Tricleidus’ svalbardensis based on new material from the Agardhfjellet Formation (middle Volgian). Norwegian Journal of Geology 92:175–186.
Knutsen, E. M., P. S. Druckenmiller, and J. H. Hurum. 2012b. Two new species of long-necked plesiosaurians (Reptilia: Sauropterygia) from the Upper Jurassic (middle Volgian) Agardhfjellet Formation of central Spitsbergen. Norwegian Journal of Geology 92:187–212.
Knutsen, E. M., P. S. Druckenmiller, and J. H. Hurum. 2012c. A new plesiosauroid (Reptilia: Sauropterygia from the Agardhfjellet Formation (middle Volgian) of central Spitsbergen, Norway. Norwegian Journal of Geology 92:213–234.
Lena L. et al. 2019. High-precision U–Pb ages in the early Tithonian to early Berriasian and implications for the numerical age of the Jurassic–Cretaceous boundary. Solid Earth, 10, 1-14, https://doi.org/10.5194/se-10-1-2019
O'Keefe F. R., Street H.P., Cavigelli J.P., Socha J.J., O'Keefe R. D. 2009. A Plesiosaur Containing an Ichthyosaur Embryo as Stomach Contents from the Sundance Formation of the Bighorn Basin, Wyoming. Journal of Vertebrate Paleontology. 29(4) 1306-1310
O’Keefe, F. R., Otero R. A., Soto-Acuna S., O’Gorman J. P., Godfrey S. J., Chatterjee S. 2017. Cranial anatomy of Morturneria seymourensis from Antarctica, and the evolution of filter feeding in plesiosaurs of the austral Late Cretaceous. Journal of Vertebrate Paleontology. 37:4, doi: 10.1080/02724634.2017.1347570.
Otero R.A., Soto-Acuña S., O'Keefe F.R. 2018. Osteology of Aristonectes quiriquinensis (Elasmosauridae, Aristonectinae) from the upper Maastrichtian of central Chile. Journal of Vertebrate Paleontology, 38:1, doi: 10.1080/02724634.2017.1408638
Rogov, M. A. 2010. New data on ammonites and stratigraphy of the Volgian stage in Spitzbergen. Stratigraphy and Geological Correlation, 18, 505–531.
Rogov MA. 2014. Infrazonal subdivision of the Volgian Stage in its type area using ammonites and correlation of the Volgian and Tithonian Stages. STRATI 2013. First International Congress on Stratigraphy. At the Cutting Edge of Stratigraphy. Cham: Springer Geology, 577–580.
Soul L., Benson R.B.J. 2017. Developmental mechanisms of macroevolutionary change in the tetrapod axis: A case study of Sauropterygia. Evolution, 71(5), 1164–1177.
Zverkov NG, Fischer V, Madzia D, Benson RBJ. 2018. Increased pliosaurid dental disparity across the Jurassic–Cretaceous transition. Palaeontology 61(6): 825–846
Zverkov NG, Prilepskaya NE. 2019. A prevalence of Arthropterygius (Ichthyosauria: Ophthalmosauridae) in the Late Jurassic—earliest Cretaceous of the Boreal Realm. PeerJ 7:e6799

The following papers not cited in the current version could also be helpful as are related to the topic of the research:

Evans M. 1999. A new reconstruction of the skull of the Callovian elasmosaurid plesiosaur Muraenosaurus leedsii Seeley. Mercian Geologist 14:191–196.

Maisch M.W. 1998. Notes on the cranial osteology of Muraenosaurus Seeley, 1874 (Sauropterygia, Jurassic), with special reference to the neurocranium and its implications for sauropterygian phylogeny. Neues Jahrbuch für Geologie und Paläontologie Abhandlungen. 207: 207–253.

---

## Round 0.2 · Minor Revisions

Dear authors,

I have accepted the reviewer’s decision of ‘minor revisions’.

I look forward to receiving your revised manuscript.

·

Basic reporting

Dear editor, dear authors,

The only point I would like to bring here is the following one, related to the sharing of data, as requested by the journal:

As my colleague reviewing the paper, I also strongly encourage the authors to find a way to provide .stl or .ply (or else) files of the volumetric renderings of the elements that have been CT-scanned. This participates to the initiative of making scientific data open and widely distributed.

Experimental design

I would like to bring two points here: one related to the topic of the paper (J/K boundary) and one related to the phylogenetic methods:

A. J/K boundary turnover:
The added caveats of the stratigraphic position of the specimen make its weird to maintain an emphasis on the J/K turnover in the paper. Now, the end of the abstract mentions that a *possibility* is *raised* that two lineages *may* have crossed the boundary. Such level of uncertainty renders this part of the MS dispensable at the new specimen is not shedding a new light on the very topic of the Jurassic/Cretaceous boundary turnover.

B. Phylogenetic methods:
The way phylogenetic methods are described (L987-1109) is still ambiguous. Here is a suggestion (please make sure it perfectly matches with what you have done):


Ophthalmothule cryostea (PMO 224.248) was scored into the data matrix of Roberts et al., (2017), which stems from the matrix from Benson and Druckenmiller (2014). Based on the results of the present study, three new characters (Characters 271-273) were created, relying on features relevant for cryptoclidids and include two cranial and one post-cranial features (See Data S1 for further discussion and description). Alternative scores and additional information on how to discern character states of individual cryptoclidid taxa are available in the supplementary information. The first new cranial character relates to the presence of an interfrontal vacuity and the second relates to the dorsal surface of the dentary. We edited a previously used postcranial character related to the fibular morphology and included in the matrix. The resulting matrix totals 273 unordered morphological characters and 76 OTUs.
The analysis was performed in TNT (V.1.5) (Goloboff and Catalano, 2016) using the parsimony ratchet, followed by a heuristic search using Tree bisection reconnection (TBR) that used the trees recovered from the parsimony rachet analysis. The parsimony ratchet analysis used 1000 ratchet iterations, with 10 random addition sequences
###
Please indicate here if you used drift and how many cycles (most of the times, people use the default value of 10)
###
Please indicate here how many trees were allowed in the memory. By default, TNT proposes of a value 100 which is may too low for any serious analysis. Many analyses nowadays go for 200.000 (the command is “hold 200000;”).
###

. All trees were kept and auto constrain turned off and all characters were equally weighted. Yunguisaurus was defined as the outgroup taxon (Cheng et al., 2006). The bremer function in TNT (V 1.5) was used to calculate Bremer support (decay index). Bootstrap resampling (1000 replications), was also performed (Fig. S.5). The scripts stats.run was used to calculate CI and RI. The strict consensus tree for Plesiosauroidea is available in supplementary information (Fig. S.14).
A posteriori time scaling of the strict consensus tree was computed in the R statistical environment and used data collected from the PBDB (Paleobiology Database, paleobiodb.org), and was completed by the literature
###
please indicate the references used, if any
###
. The Datephylo function from the R package strap (Bell and Lloyd, 2014), was utilised to form the time scaled tree (See supplementary information for data).

###
Please indicate the way the tree was timescaled (i.e. how the length of internal branches was computed): equal? Mbl? Modified Hedman?
###

Validity of the findings

A comment on evolutionary drivers:

The section (L1108 to 1130) appears unclear and I find it difficult to follow your thinking process. Here are some points/thoughts that I hope will help you regarding your interpretation of the cranial shape of Ophthalmothule :

In said section, you seem to oppose the ‘mean’ of evolving larger eyes (through the evolutionary process of paedomorphy) versus the possible driver (increase visual acuity) and versus the possible relative changes due to allometry and the integration of the skull (the reduction in relative size of other cranial parts). All of these can actually be combined to explain a phenotype.

In lines 1123 to 1125, you seem to suggest that the skull is fixed in total length and that any change in one part (e.g. larger eyes) necessarily result in shortening of the others. However, skulls are never 100% integrated and it is well known in plesiosaurians and marine reptiles in general that some parts can be extended without necessarily reducing the others (e.g. the longer snouts and larger eyes in ichthyosaurs).

An important point to add here is that mechanical advantages are computed as ratios and are thus dimensionless while visual acuity is based on the *absolute* size of the eye (notably the diameter of the dilatated pupil). Hence, an animal with a shorter rostrum and similarly sized eyes than its ancestor would have a different mechanical advantage and relatively larger eyes, but no change in actual visual acuity at all. Comparing the ancestral condition to that of Ophthalmothule might be the key to highlight the changes going in that particular lineage.

Additional comments

Line-by-line minor comments:

L25: Could you please add “ and species” after “genus”.

L26: double space before “The holotype”

L28: double space before “observed”

L28: “incorporating” instead of “incorporting”

L40 and throughout. “Plesiosaurian” and “plesiosaur” are used interchangeably throughout the text. Could you make it more consistent?

L893: “Andrews” instead of “Andres”

L1014: could state whether these are ambiguous (i.e. ‘local’, homoplasic) or non-ambiguous (i.e. ‘global’, non-homoplasic) synapomorphies?

L1134, L1136: It is best not to not start a sentence with an abbreviated term (here “O.”)


Best wishes,

Valentin Fischer

·

Basic reporting

As I reviewed the MS before and suggested minor revisions, I will not follow the recommended scheme of review this time. All my comments are in "comments for the author" section.

I'm happy with the changes the Authors did.

There are several typos I found (listed in comments for the author section).

My recommendation would be rather to accept the paper as is. However, as I added some additional comments, I put a tick near "minor revision" again so that authors have time to correct typos and, if they consider it necessary (hopefully not), to respond to my comments.

Experimental design

no comments

Validity of the findings

no comments

Additional comments

I feel the need to comment on some of the Authors' rebuttals. These comments are given below along with other line-by-line corrections.

Line 25
“representing a new genus Ophthalmothule cryostea gen et sp. nov.,”

Either write “taxon” or “genus and species” as in the current form you wrote genus but then give a name of the new species.

Line 31
“incorporting”

incorporAting?

Line 169
“basioccipital tubera broad and flattened”

As I asked in the previous round, please add how these are broad mediolaterally or anteroposteriorly?
Surprisingly in your rebuttal, you wrote that corrected this, but actually not.

Line 199
“Kimmerosaurus langhami”

As I noted in the previous round, mid-posterior cervicals are not known for this taxon, at least in published works, so please either add “pers. obs.” and reference to the undescribed specimen in which you observed this trait (with the number), or delete the mention of this taxon. This is not critical at all. As I spent a couple of weeks studying the collections of the NHMUK, I can say that, as far as I know, there is only one specimen of Kimmerosaurus with cervical vertebra preserved – NHMUK R 10042, described by Brown, Milner & Taylor (1986). This specimen has 5! postaxial cervical centra preserved. You are welcome to share your observations on the undescribed specimens from the Etches collection, but please make this clear to readers. Still, I do not understand why to write: “Response: There are anterior-mid cervical available for Kimmerosaurus at the NHM in London. There are also more posterior vertebrae present in the Etches collection.”
Do you mean any specimen from the NHMUK that has more than 5 postaxial vertebrae preserved? So please give a reference as this is not formally described to my knowledge, or you imply that Kimmerosaurus had 15-20 cervical vertebrae (so that 7th vertebra could be considered "anterior-mid cervical")?

Line 201-202

"L 199 “short and reduced”
I would not be so sure. It seems more like broken and lost than reduced.
“Response: As I (A Roberts) personally excavated and prepared the specimen, I can personally stand for that the dorsal process of the scapula is indeed low and not broken. On the right side, there is some damage due to the overlying vertebrae and ribs. However, the left side is complete, although cracked due to congelifraction.”

Please remember that as a reviewer I’m trying to help you not to make a mistake and/or misinterpretation. Personal excavation of the specimen does not guarantee the original completeness of the specimen and that the broken part was not displaced during taphonomy. I cannot insist on the suggestion but hope that your further research of plesiosaurian materials will help your better understanding of the morphology of their pectoral girdle. As I (N Zverkov) personally re-examined the plesiosaur and ichthyosaur materials from Svalbard that you and your colleagues have previously described, I know well the problems with preservation and taphonomic tricks in this stuff – e.g. the case with the right scapula of the holotype of plesiosaur Djupedalia engeri that was excavated as three disarticulated and scattered fragments and subsequently described as clavicles and interclavicle and even added to the diagnosis of the new taxon (Knutsen et al., 2012). Their broken edges were identified as facets, but when all three parts are putten together they form a complete right scapula instead:
https://www.dropbox.com/sh/gnjbafrwmcgvdli/AAArYyNwYK5n6_XgkJ0tlwi5a?dl=0
As far as I know, you were informed about this my "discovery" by J Hurum right during my visit. I haven't checked your coding for this particular case, but hope that you considered this in the current contribution.
This is just one example of how taphonomy can bias interpretations and further work. With your specific Svalbard material, such things should be always kept in mind.
Please beware the misinterpretations and better examine the morphology of scapulae in other cryptoclidids including Muraenosaurus, Tricleidus and Cryptoclidus (e.g. Andrews, 1910).


Line 210
“The preserved skeleton of PMO 224.248 is well-preserved and fully articulated”

Consider rewording, e.g. deleting the first “preserved”.

“ L 561-562
“possibly due to taphonomy” - you can say the same regarding PMO 224.248. Based on my observations on this nicely preserved specimen it is more likely a natural condition than a taphonomic distortion.
“it appears absent on the holotype” - the mandible of the holotype is disarticulated. The dentaries are nearly completely preserved and fused. their posterior ends are mediolaterally wide. It is possible that this could be due to taphonomy, but I would rather suggest that this is a natural condition further supporting your observation of this morphology in the referred specimen.
Photograph of dentaries of the holotype - https://www.dropbox.com/s/g40mop6l28nuyka/IMG_5879.JPG?dl=0
Response: Corrected and edited to support this”

I see no correction that agrees with the observations. The feature is present in the holotype specimen. I have no idea why you are still insisting on your interpretation even after getting photographs of the specimen – leaving this at your conscience.


“L 603 “ridging is most prominent on the labial side”
Could you provide close-up photographs of the enamel ornamentation as supplemental files, please? This interpretation sounds confusing to me and it would be nice to have pictures for agreeing that your description is correct.
Response: Unfortunately, I do not have access to the specimen currently and cannot get more detailed photos of the dentition. Although we do believe the current images suffice as supplement to the description. Regarding the D-shaped cross-section in Kimmerosaurus, this was an error on our part and has now been fixed in text. ”

That’s unfortunate, that from the five authors of the paper only the lading author is capable of taking additional pictures, but cannot get them as well… Hopefully, you will be able to provide these in your further works.


Line 893
“Andres”
Andrews

Line 930
“mansellii”
manselii

“L914-915
Please delete the sentence as it is speculation
(i) - there are two elements in 'P.' manselii, and one preserved in PMO 214.248 - doesn't mean there was no second or even third pae.
(ii) - the posterodistal edge of the humerus in 'P.' manselii is broken and lost, thus its morphology is unknown
Response: Regarding this sentence “This is somewhat different from “Plesiosaurus” manselii, where the postaxial elements occupy the entire distal postaxial margin (Hulke, 1870).” We have edited it to make it clear that this evidence is suggested from the publication and not from the specimen itself.”

In case you want to support your statement by the reference only, please see the original publication carefully and pay attention that the posterodistal portion of the humerus was missing during the original description (Hulke, 1870: pl XLI, fig 3). To leave this citation you should write something like: "according to the reconstruction of Hulke (1870)".


“L 945 “tibiale” and “astragalus”
if you have astragalus, you can’t have the tibiale the same time. either interpret it as a centrale, or apply a traditional interpretation with the intermedium (not astragalus) and tibiale.
I had no time for publishing this, but discussed at SATLW-2017 - https://www.researchgate.net/publication/319454783_MESOPODIAL_ELEMENTS_IN_HINDLIMBS_OF_ICHTHYOSAURS_AND_PLESIOSAURS_CONTROVERSIAL_INTERPRETATIONS_AND_POSSIBLE_SOLUTION
Response: As far as I am aware it is correct terminology to use astragalus in the hind limb and intermedium in the fore limb. As such, I have let this stand.”

This is not critical and you can interpret the element in any way you want to. Just for accuracy of your further work I recommend to read something about limb evolution and development in tetrapods – this is not only interesting but also very useful. Regarding the interpretations in limbs of sauropterygians you can read e.g. Caldwell, 1997 (Limb osteology and ossification patterns in Cryptoclidus….)


Best wishes

Nikolay

---

## Round 0.3 · accepted · Accept

Dear authors,

After reading your response to reviewers .doc I have accepted your manuscript for publication.

PeerJ production staff will soon be in touch to discuss your proofs.

Once again, thank you for choosing PeerJ as your publishing venue, and I hope you use us again in the future.